# FlowHMM: Flow-based continuous hidden Markov models

**Paweł Lorek**
Mathematical Institute
University of Wrocław
Tooploox
pawel.lorek@math.uni.wroc.pl

**Rafał Nowak**
Institute of Computer Science
University of Wrocław
Tooploox
rafal.nowak@cs.uni.wroc.pl

**Tomasz Trzciński**
Warsaw University of Technology
Jagiellonian University of Cracow
Tooploox, IDEAS NCBR
tomasz.trzcinski@pw.edu.pl

**Maciej Zięba**
Department of Artificial Intelligence
Wrocław University of Science and Technology
Tooploox
maciej.zieba@pwr.edu.pl

## Abstract

Continuous hidden Markov models (HMMs) assume that observations are generated from a mixture of Gaussian densities, limiting their ability to model more complex distributions. In this work, we address this shortcoming and propose novel continuous HMM models, dubbed FlowHMMs, that enable learning general continuous observation densities without constraining them to follow a Gaussian distribution or their mixtures. To that end, we leverage deep flow-based architectures that model complex, non-Gaussian functions and propose two variants of training a FlowHMM model. The first one, based on gradient-based technique, can be applied directly to continuous multidimensional data, yet its application to larger data sequences remains computationally expensive. Therefore, we also present a second approach to training our FlowHMM that relies on the co-occurrence matrix of discretized observations and considers the joint distribution of pairs of co-observed values, hence rendering the training time independent of the training sequence length. As a result, we obtain a model that can be flexibly adapted to the characteristics and dimensionality of the data. We perform a variety of experiments in which we compare both training strategies with a baseline of Gaussian mixture models. We show, that in terms of quality of the recovered probability distribution, accuracy of prediction of hidden states, and likelihood of unseen data, our approach outperforms the standard Gaussian methods.

## 1 Introduction

Hidden Markov models (HMMs) are a standard tool in modeling and analysis of time series data. Although structurally simple, they have been successfully applied in a wide variety of applications, ranging from finance [1], speech recognition [2] to computational biology [3] and climate modeling [4].

HMMs are capable of solving complex problems, but their adoption is limited due to several reasons. First, the training process of HMMs typically relies on the Baum-Welch algorithm [5] — a particular case of expectation-maximization (EM) method, which offers a relatively slow convergence to a local maximum. To reduce this burden, several discrete HHMs introduce the so-called co-occurrence matrix that aggregates information about the probability of jointly observed values in the chain and estimate

36th Conference on Neural Information Processing Systems (NeurIPS 2022).

it together with the parameters of the whole model [6, 7, 8, 9, 10]. Although the co-occurrence matrix is computed only once and then used to significantly reduce convergence time during optimization, its application is strictly limited to discrete HMMs and cannot be easily generalized to their continuous variants.

Secondly, continuous HMM models are restricted to follow standard parametrized distributions when modeling the observations. Most of them use either Gaussians or other parametric families of distributions [11, 12], while the others rely on the mixtures of Gaussians or apply semiparametric distribution estimation [13, 14]. As a result, the existing HMMs have limited ability to model complex observations that do not follow the distributions mentioned above. This, in turn, hinders their application in real-life use cases, *e.g.*, in human action recognition [15].

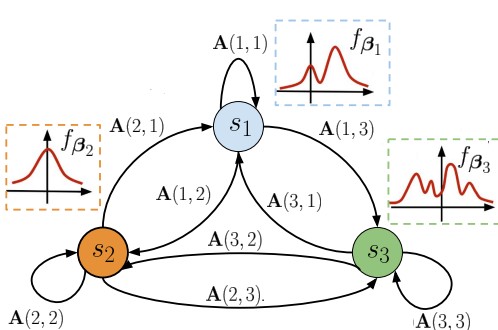

Figure 1: The concept of FlowHMM for $L = 3$ states and transition matrix $\mathbf{A}$. Each emission distribution characterised by density $f_{\boldsymbol{\beta}_l(\cdot)}$ is modeled using a separate flow component. Thanks to this, they can adjust to complex, non-Gaussian distributions.

In this work, we address the shortcomings of the existing HMMs and introduce FlowHMM, a novel continuous hidden Markov model that learns a general continuous distribution of observations by exploiting the properties of flow-based models [16]. The core idea of our approach is to model the distributions of the observations with normalizing flows, instead of Gaussians or their mixtures. Flows map a simple Gaussian prior to more complex distributions using a parametrized neural network. This formulation enables flow-based models to overpass parametric models in their flexibility to handle data samples of unknown distributions. Moreover, flow-based models naturally extend to a multimodal setting, which effectively renders obsolete the tedious process of tuning the number of hidden states.

Practically, training the model with flow-based components requires a gradient-based optimization. To achieve that, we propose two variants of training the FlowHMM model. The first approach is based on maximum likelihood (ML) technique and can be applied directly to continuous multidimensional data. In the second method, we discretize the continuous values during training and leverage the co-occurrence matrix. As a result, we provide an end-to-end training procedure that jointly optimizes the flow-based component parameters and the co-occurrence matrix using standard gradient-based techniques. Since the model is trained in a discretized form, the optimization process is simple and more efficient than the competing HMM training procedures, while during inference, we can still use a continuous version of our FlowHMM model.

To summarize, the *main contribution of our paper* is a novel continuous HMM method dubbed FlowHMM with two alternative training scenarios, capable of modeling complex multimodal distributions of observations without constraining them to follow Gaussians. Not only does it outperform the competing approaches, but it also increases the efficiency of the required optimization procedure.

## 2 Background

### 2.1 Hidden Markov models

Let $\{X_k\}_{k\geq 0}$ be an ergodic, time homogeneous Markov chain over hidden states $\mathcal{S} = \{s_1, \ldots, s_L\}$ with a stationary distribution $\boldsymbol{\pi} = (\pi_1, \ldots, \pi_L)$, and a transition matrix $\mathbf{A}$, *i.e.*, for any $k$ we have $\mathbb{P}(X_{k+1} = s_j | X_k = s_i) = \mathbf{A}(i, j)$. Let $\{Y_k\}_{k\geq 0}$ be a sequence of random variables taking values in $\mathcal{V}$ (the *observation space*), which can be continuous or discrete. Let us fix some time horizon $T$. Random variables $Y_{0:T} = (Y_0, \ldots, Y_T)$ are independent conditionally on the state sequence $X_{0:T} = (X_0, \ldots, X_T)$, *i.e.*:

$$p(Y_{0:T} = y_{0:T} | X_{0:T} = x_{0:T}) = \prod_{k=0}^{T} p(y_k | x_k), \qquad (1)$$

where $p(y|x) \equiv \mathbb{P}(Y = y|X = x)$ represents the so-called *emission probabilities* for discrete observations. We will shortly write $p(y_{0:T})$ for $p(Y_{0:T} = y_{0:T})$ and similarly *e.g.*, $p(y_{0:T}|x_{0:T})$ for $p(Y_{0:T} = y_{0:T}|X_{0:T} = x_{0:T})$ to simplify the notation. Considering the continuous case, $p(y|x)$ represents the conditional emission density function for a given state $x$. For a Gaussian Mixture HMM (which we simply call Gaussian HMM) we assume that $p(y|x) = \sum_{k=1}^{K} \alpha_{x,k} \mathcal{N}(y; \mu_{x,k}, \sigma_{x,k}^2)$, where $\alpha_{x,\cdot}$ is a distribution on $\{1, \ldots, K\}$. The values of parameters $\{\alpha_{x,k}, \mu_{x,k}, \sigma_{x,k}^2\}_{k=1}^{K}$ are determined by the conditioning value $x$. Note that we assume that each mixture has the same number of components $K$. For $K = 1$ we have a classic Gaussian HMM.

HMM model can be parametrized by $\boldsymbol{\theta} = \{\boldsymbol{\pi}, \mathbf{A}, \boldsymbol{\beta}\}$, where $\boldsymbol{\beta} = \{\boldsymbol{\beta}_1, \ldots \boldsymbol{\beta}_L\}$, and $\boldsymbol{\beta}_l$ stays behind the parameters of $p(y|l)$, for a given state $l$. For discrete case, $\boldsymbol{\beta}_l$ represents the parameters for categorical distribution, while for a Gaussian HMM we have $\boldsymbol{\beta}_l = \{\alpha_{l,k}, \mu_{l,k}, \sigma_{l,k}^2\}_{k=1}^{K}$. The probability of observing the sequence of observations $y_{0:T}$ can be expressed as:

$$p(y_{0:T}; \boldsymbol{\theta}) = \sum_{x_{0:T} \in \mathcal{S}^{T+1}} p(y_{0:T}|x_{0:T}; \boldsymbol{\beta}) p(x_{0:T}; \mathbf{A}, \boldsymbol{\pi}), \tag{2}$$

where $p(y_{0:T}|x_{0:T}; \boldsymbol{\beta})$ is given by Eq. (1), and $p(x_{0:T}; \mathbf{A}, \boldsymbol{\pi})$ can be expressed as:

$$p(x_{0:T}; \mathbf{A}, \boldsymbol{\pi}) = \mathbb{P}(x_0) \prod_{k=1}^{T} \mathbb{P}(x_k|x_{k-1}) = \pi_{x_0} \prod_{k=1}^{T} \mathbf{A}(x_{k-1}, x_k). \tag{3}$$

## 2.2 Normalizing Flows

Normalizing flows [16] are generative models that can be efficiently trained via direct likelihood estimation thanks to the application of the change-of-variable formula. Practically, they utilize sequence of parametric and invertible transformations: $y = h_n \circ \cdots \circ h_1(z)$. The goal of the transformation is to map $z$ from the known, normal distribution $\mathcal{N}(z; \mathbf{0}, \mathbf{I})$ to the more complex distribution described by a density function $f(y)$ from the observation domain. The log-probability for $y$ can be expressed as:

$$\log f(y) = \log \mathcal{N}(z; \mathbf{0}, \mathbf{I}) - \sum_{n=1}^{N} \log \left| \det \frac{\partial h_n}{\partial z_{n-1}} \right|. \tag{4}$$

One of the main challenges while designing the normalizing flows is selection of a proper form of transformation functions $h_n$. The sequence of discrete transformations can be replaced by continuous equivalent by application of Continuous Normalizing Flows (CNFs) [17], where the aim is to solve differential equation of the form $\frac{dz}{dt} = g_{\boldsymbol{\beta}}(z(t), t)$, where $g_{\boldsymbol{\beta}}(z(t), t)$ represents the function of dynamics, described by parameters $\boldsymbol{\beta}$. Our goal is to find a solution of the equation in $t_1$, $y := z(t_1)$, assuming the given initial state $z := z(t_0)$ with a known prior. The transformation function $h_{\boldsymbol{\beta}}$ and its inverse are defined as defined as:

$$y = h_{\boldsymbol{\beta}}(z) = z + \int_{t_0}^{t_1} g_{\boldsymbol{\beta}}(z(t), t) \, dt, \quad h_{\boldsymbol{\beta}}^{-1}(y) = y - \int_{t_0}^{t_1} g_{\boldsymbol{\beta}}(z(t), t) \, dt. \tag{5}$$

The log-probability of $y$ can be computed by (where $h_{\boldsymbol{\beta}}^{-1}(y) = z$):

$$\log f_{\boldsymbol{\beta}}(y) = \log \mathcal{N}(h_{\boldsymbol{\beta}}^{-1}(y); \mathbf{0}, \mathbf{I}) - \int_{t_0}^{t_1} \frac{d g_{\boldsymbol{\beta}}(z(t), t)}{d z(t)} \, dt. \tag{6}$$

The choice of using CNFs is motivated by the fact, that we are focused on modeling distributions of one or low-dimensional data. CNFs were successfully applied in such models as NGGP [18], PointFlow [19], or StyleFlow [20], where the dimensionality and characteristic of data are similar. It is somehow confirmed by the empirical results provided by the authors of FFJORD ([17] Table 2), the proposed approach performs better than discrete flows like RealNVP [21] or Glow [22] for low-dimensional data in terms of normalized log-likelihood. Such models were used in [23]. Moreover, flows that use coupling layers (RealNVP, Glow) and autoregressive flows (MAF [24]) do not make sense for 1D data. While operating on 1D data, we do not care about simplifying the estimation of the Jacobian, and any invertible differentiable transformation can be applied. At the same time, we need a complex, well-parameterized transformation that is delivered by a dynamic function of CNF.

## 3 FlowHMM

In this section we introduce FlowHMMs - HMM variants of continuous flow capable to model the observations using complex, non-Gaussian distributions. The idea behind this approach is to model each of conditional densities $p(y|x = s_l) = f_{\beta_l}(y)$, for each of the considered states, $s_l \in S$, using a separate CNF module. In practice, $p(y|x = s_l)$ can be calculated using formula given by Eq. (6) with the parameters $\beta_l$ dedicated for the state $s_l$. The idea of our approach is illustrated in Fig. 1.

In FlowHMM we assume that $\{X_k\}$ is stationary, *i.e.*, the stationary distribution $\pi$ is also its initial distribution. In such a case, instead of using $\mathbf{A}$, the model can be equivalently represented by the state joint probabilities:

$$\mathbf{S}(i, j) = \mathbb{P}(X_k = s_i, X_{k+1} = s_j) = \mathbb{P}(X_{k+1} = s_j | X_k = s_i)\mathbb{P}(X_k = s_i) = \mathbf{A}(i, j)\pi_i. \quad (7)$$

Note that $\pi_i$ can be computed as $\sum_j \mathbf{S}(i, j)$, what can be written as $\pi_i = \mathbf{1}_i \mathbf{S} \mathbf{1}^T$, where $\mathbf{1}$ denotes a vectors consisting of ones and $\mathbf{1}_i$ consists of 1 on position $i$ and zeros otherwise. For such a formulation, the FlowHMM model can be parametrized by $\boldsymbol{\theta} = \{\mathbf{S}, \beta_1, \ldots \beta_L\}$.

We propose two variants of FlowHMM models, which differ mainly in a training process: *gradient-based model* $\mathcal{F}^{\mathrm{ML}}$ which is trained with an maximum likelihood approach directly on continuous data, and *co-occurrence matrix-based model* $\mathcal{F}^{\mathbf{Q}}$, that utilizes co-occurrence matrix and is trained in end-to-end setting using discretized sequences. The first of the proposed methods does not require discretisation step, but is costly and ineffective for larger sequences. $\mathcal{F}^{\mathbf{Q}}$ eliminates that problem, and makes the training time independent of the length of the training sequence (only estimating empirical co-occurrence matrix depends on $T$, but the time is marginal). In upcoming sections we are going to introduce both models in more details.

### 3.1 Training FlowHMM $\mathcal{F}^{\mathrm{ML}}$ model

Given a sequence of observations $y_{0:T} = (y_0, \ldots, y_T)$, the model is trained by optimizing the log-likelihood $\log p(y_{0:T}; \boldsymbol{\theta})$ (see Eq. (2)), where we aim at finding the optimal values of $\mathbf{S}^*$ (and thus $\mathbf{A}^*$ and $\pi^*$) and parameters of flow models $\beta_l^*$, such that $\boldsymbol{\theta}^* = \{\mathbf{S}^*, \beta_1^*, \ldots, \beta_L^*\} = \arg\max_{\boldsymbol{\theta}} \log p(y_{0:T}; \boldsymbol{\theta})$. In order to satisfy the constraints on $\mathbf{S}$ (*i.e.*, that $\mathbf{1S1}^T = 1$ and each entry is non-negative, what we denote by $\mathbf{S} \geq \mathbf{0}$) we parametrize the matrix $\mathbf{S}$ by a real-valued matrix $\tilde{\mathbf{S}}$ also of size $L \times L$, using the following softmax function:

$$\mathbf{S}(s_1, s_2) = \frac{\exp(\tilde{\mathbf{S}}(s_1, s_2))}{\sum_{s_i, s_j} \exp(\tilde{\mathbf{S}}(s_i, s_j))}. \quad (8)$$

Thanks to that representation, we train the $\mathbf{S}$ and $\beta_l$'s iteratively with gradient-based approach, by maximizing the incomplete log-likelihood $\log p(y_{0:T}; \boldsymbol{\theta})$, calculated directly using the forward algorithm.

### 3.2 Training FlowHMM $\mathcal{F}^{\mathbf{Q}}$ model

Considering $\mathcal{F}^{\mathbf{Q}}$ model we introduce the *co-occurrence matrix* that represents the joint distribution of two consecutive observations: $\mathbf{Q}(y_1, y_2) = p(Y_{k+1} = y_2, Y_k = y_1)$. The matrix represents a categorical distribution, $(i, j)$-th entry represents the probability of observing a pair of states $(v_i, v_j)$ at some fixed two consecutive steps $k$ and $k + 1$. Note that it is independent of $k$, since throughout the paper we assume that underlying Markov chain on hidden states is stationary, thus a bivariate distribution of $(X_k, X_{k+1})$ is independent of $k$. The matrix can be rewritten as:

$$\mathbf{Q}(y_1, y_2) = \sum_{s_i, s_j \in \mathcal{S}} p(y_1|s_i)\mathbf{S}(i, j)p(y_2|s_j). \quad (9)$$

Assuming the discrete set of observations, $\mathcal{V} = \{v_1, \ldots, v_M\}$, it can be further expressed as $\mathbf{Q} = \mathbf{P}^T \mathbf{S} \mathbf{P}$, where $\mathbf{P}$ collects probabilities of all possible observations at each hidden state in a matrix $\mathbf{P}(s_i, v_j) = p(v_j|s_i)$. In this case there are $M^2$ possible observation pairs $(y_1, y_2) \in \mathcal{V} \times \mathcal{V}$. Note that matrices $\mathbf{Q}, \mathbf{S}, \mathbf{P}$ are of sizes $M \times M$, $L \times L$ and $L \times M$, respectively. Moreover, we have $\sum_{v_i, v_j \in \mathcal{V}} \mathbf{Q}(v_i, v_j) = \sum_{s_i, s_j} \mathbf{S}(s_i, s_j) = 1$ and $\sum_{v \in \mathcal{V}} \mathbf{P}(s_i, v) = 1$ for all $s_i \in \mathcal{S}$. In a matrix

form, these can be written as $\mathbf{1}\mathbf{Q}\mathbf{1}^T = \mathbf{1}\mathbf{S}\mathbf{1}^T = 1$ and $\mathbf{P}\mathbf{1}^T = \mathbf{1}^T$ (note that $\mathbf{1}$ must be of an appropriate size). Given observations $y_{0:T}$ the matrix $\mathbf{Q}$ can be empirically estimated by:

$$\hat{\mathbf{Q}}(v_i, v_j) = \frac{1}{T} \sum_{k=0}^{T-1} \mathbb{I}(y_k = v_i)\mathbb{I}(y_{k+1} = v_j), \tag{10}$$

for all pairs $(v_i, v_j) \in \mathcal{V} \times \mathcal{V}$. The problem of training the HMM is to find such parameters (matrices $\mathbf{S}$ and $\mathbf{P}$) so that the co-occurrence matrix $\mathbf{Q}$ is close (in some sense) to the empirical co-occurrence matrix $\hat{\mathbf{Q}}$. We can formulate the problem as (for some distance $\mathrm{dist}$)

$$\min_{\substack{\mathbf{P} \in \mathbb{R}^{N \times M}, \\ \mathbf{S} \in \mathbb{R}^{L \times L}}} \mathrm{dist}(\hat{\mathbf{Q}}, \mathbf{P}^T \mathbf{S} \mathbf{P}), \qquad \text{subject to} \quad \mathbf{1}\mathbf{S}\mathbf{1}^T = 1, \mathbf{P}\mathbf{1}^T = \mathbf{1}^T, \mathbf{P} \geq \mathbf{0}, \mathbf{S} \geq \mathbf{0}. \tag{11}$$

This problem formulation has a couple of advantages compared to standard likelihood-based optimization. First, the empirical co-occurrence matrix $\hat{\mathbf{Q}}$ is independent of the sequence length $T$. Second, the given objective can be easily optimized using gradient-based techniques. On the other hand, the constraints for matrices $\mathbf{P}$ and $\mathbf{S}$ should be satisfied. The set $\mathcal{V}$ is discrete, while we aim to design the model for continuous observations. In order to satisfy the constraints for matrix $\mathbf{S}$ we use the representation given by Eq. (8). In order to represent matrix $\mathbf{P}$ we use Flow-based emission probabilities.

**Flow-based emission probabilities.** With each hidden state $l$ we associate the flow model described by a density function $f_{\boldsymbol{\beta}_l}$, that can be calculated from Eq. (6), where $\boldsymbol{\beta}_l$ is a set of trainable parameters of the flow. We construct $\mathbf{P}$ based on these models (in such a case $\mathbf{P} \equiv \mathbf{P}_{\boldsymbol{\beta}}$). The $i$-th row of the matrix is a density $f_{\boldsymbol{\beta}_i}(\cdot)$ evaluated at $v_1, \ldots, v_M$ and normalized as follows:

$$\mathbf{P}_{\boldsymbol{\beta}}(s_i, v_j) = \frac{f_{\boldsymbol{\beta}_i}(v_j)}{\sum_{k=1}^{M} f_{\boldsymbol{\beta}_i}(v_k)}. \tag{12}$$

Similarly, the optimization problem (11) becomes:

$$\min_{\boldsymbol{\beta}, \mathbf{S} \in \mathbb{R}^{L \times L}} \mathrm{dist}(\hat{\mathbf{Q}}, \mathbf{P}_{\boldsymbol{\beta}}^T \mathbf{S} \mathbf{P}_{\boldsymbol{\beta}}). \tag{13}$$

As a distance function, we propose to use *Kullback–Leibler divergence*,

$$\min_{\boldsymbol{\beta}, \mathbf{S} \in \mathbb{R}^{L \times L}} \mathcal{L}, \quad \mathcal{L} = \sum_{i,j} \mathbf{Q}_{\boldsymbol{\beta}}(i,j) \log \frac{\mathbf{Q}_{\boldsymbol{\beta}}(i,j)}{\hat{\mathbf{Q}}(i,j)}, \tag{14}$$

where $\mathbf{Q}_{\boldsymbol{\beta}} = \mathbf{P}_{\boldsymbol{\beta}}^T \mathbf{S} \mathbf{P}_{\boldsymbol{\beta}}$. We postulate to apply divergence instead to $L^2$ distance used *e.g.*, in [7]., because it is more natural measure for comparing two distributions. We observed during empirical evaluation, that using this divergence gives better stability and convergence of the training process.

The final training procedure of FlowHMM $\mathcal{F}^{\mathbf{Q}}$ model is as follows. Let $y_{0:T}^{\mathrm{train}}$ be the training set and let $y_{0:T'}^{\mathrm{test}}$ be the test set. For the fixed $M$ we construct the grid $\Gamma = (\gamma_1, \ldots, \gamma_M)$ using one of the approaches described in A.1. Next, we create the discretized training data $y_{0:T}^{\mathrm{train},\Gamma}$, where $y_i^{\mathrm{train},\Gamma} = \arg\min_{\gamma \in \Gamma} ||\gamma - y_i^{\mathrm{train}}||^2$. The values of the grid represent further the set $\mathcal{V}$, $\mathcal{V} = \Gamma$. Next, we calculate the empirical co-occurrence matrix $\hat{\mathbf{Q}}$:

$$\hat{\mathbf{Q}}(v_i, v_j) = \frac{1}{T} \sum_{k=0}^{T-1} \mathbb{I}(y_k^{\mathrm{train},\Gamma} = v_i)\mathbb{I}(y_{k+1}^{\mathrm{train},\Gamma} = v_j), \tag{15}$$

and matrices $\mathbf{S}$ and $\mathbf{P}_{\boldsymbol{\beta}}$ (using Eqs. (8), (12)) that are further used to calculate the loss given by Eq. (14). Practically, we add Gaussian perturbations to the grid values, while calculating the matrix $\mathbf{P}$:

$$\mathbf{P}_{\boldsymbol{\beta}}(s_i, v_j) = \frac{f_{\boldsymbol{\beta}_i}(v_j + \epsilon)}{\sum_{k=1}^{M} f_{\boldsymbol{\beta}_i}(v_k + \epsilon)}, \tag{16}$$

where $\epsilon$ is a random sample from $\mathcal{N}(0, \sigma_{noise}^2)$, and $\sigma_{noise}^2$ is a hyperparameter of the method.

This is one of the standard tricks applied while training normalizing flows, that imitates the situation where we have an access to an infinite number of training examples. These perturbations prevent from overfitting of the flow-based components caused by observing the same grid while training. Next, all of the parameters of FlowHMM are updated with gradient based approach. The procedure is repeated until convergence. The procedures of training is presented in Algorithm 1.

## 3.3 Inference with FlowHMM

As we postulated before, our model is designed for continuous problems, due to the fact, that both of the training techniques return estimated $\mathbf{S}$, and the parameters, that represent emission distributions for each of the states. Thus, during the inference stage, we simply calculate $\mathbf{A}$ and $\boldsymbol{\pi}$ from $\mathbf{S}$ and apply forward procedure on the test data $y_{0:T'}^{\text{test}}$ to calculate $p(y_{0:T'}^{\text{test}}; \boldsymbol{\theta})$. We can also determine the hidden state values using the *Viterbi* procedure. Concluding, FlowHMM can be used in the same applications as standard continuous HMM models, but with no restrictions to emission distributions.

---

**Algorithm 1** Training using $\mathcal{F}^{\mathbf{Q}}$ technique

---

**Require:** $\hat{\mathbf{Q}}$: — empirical co-occurrence matrix from Eq. (10).
**Parameters:** $\boldsymbol{\beta} = \{\boldsymbol{\beta}_1, \ldots, \boldsymbol{\beta}_L\}$ — flow parameters, $\tilde{\mathbf{S}}$ parameters representing un-normalised co-occurrence matrix.
**Hyperparameters:** $L$ - number of hidden states, $\alpha$- step size, noise variance $\sigma_{noise}^2$.

---

1: **function** TRAIN($\hat{\mathbf{Q}}, L, \alpha$)
2:     Initialize $\boldsymbol{\beta}$, and $\tilde{\mathbf{S}}$.
3:     **while** not convergent **do**
4:         Calculate $\mathbf{S}$ from Eq. (8). and $\mathbf{P}_\beta$ from Eq. (16).
5:         Calculate loss function $\mathcal{L}$ from Eq. (14) using $\hat{\mathbf{Q}}$.
6:         $\tilde{\mathbf{S}} \leftarrow \tilde{\mathbf{S}} - \alpha\nabla_{\tilde{\mathbf{S}}}\mathcal{L}$
7:         **for each** $l \in \{1, \ldots, L\}$ **do**
8:             $\boldsymbol{\beta}_l \leftarrow \boldsymbol{\beta}_l - \alpha\nabla_{\boldsymbol{\beta}_l}\mathcal{L}$
9:         **end for**
10:    **end while**
11:    **return** $\boldsymbol{\beta}_1, \ldots, \boldsymbol{\beta}_L, \tilde{\mathbf{S}}$
12: **end function**

---

**FlowHMM for multidimensional observations.** Our approach can be easily extended to multidimensional observations, *i.e.*, to the case where $y_{0:T}^{\text{train}}$ are from $\mathbb{R}^d$. For such a case, the $\mathcal{F}^{\text{ML}}$ model is straightforward, let us focus on a description of $\mathcal{F}^{\mathbf{Q}}$. On one hand the extension is straightforward: we construct some grid $\Gamma = (\gamma_1, \ldots, \gamma_M)$ of $d$-dimensional points, we create discretized training set $y_{0:T}^{\text{train},\Gamma}$ and the empirical co-occurrence matrix $\hat{\mathbf{Q}}$ from (15), afterwards we compute $\mathbf{P}_\beta$ from Eq. (16) and the gradient $\nabla_{\boldsymbol{\beta}_l}\mathcal{L}$ of $\mathcal{L}$ given in (14). In other words we proceed with an Algorithm 1. Note that CNFs are well-suited for multivariate observations. On the other hand, there is a challenging aspect of training $\mathcal{F}^{\mathbf{Q}}$ model for multidimensional data: an effective discretization technique. We elaborate on that in A.1 proposing a new grid search method. We suggest using $\mathcal{F}^{\mathbf{Q}}$ model for lower-dimensional data, especially with long observation sequence (recall, the matrix $\mathbf{Q}$ is computed once) – in such a case $\mathcal{F}^{\text{ML}}$ will usually be very slow, each epoch loops through a whole observation sequence. On the other hand, for $\mathcal{F}^{\text{ML}}$ is better suited for high-dimensional data and short observation sequence – for longer observation sequences, in order to shorten the execution time, we apply small *trick*, at each epoch we sample a subsequence of fixed length, *e.g.*, $10^3$. There is a trade-off between execution time and performance, see Table 7 in A.7.

## 4 Related work

One of the most popular way to train HMM models is the Baum-Welch algorithm (a name for EM applied to HMMs). However, one of (several) drawbacks of using Baum-Welch algorithm for training is that it is prohibitively slow for long sequences. The complexity of the forward-backward algorithm (which must be run for each epoch) is $\mathcal{O}(N^2T)$, where $N$ is the number of hidden states and $T$ is the length of the observation sequence. In order to eliminate this issue, the authors [6] propose to use pairwise co-occurrence probabilities (also higher order statistics are also discussed there) and *prove* that it is possible to recover the structure of an HMM based only on co-occurrences. An interesting extension (both, of pairs and triplets) was considered in [25]. Authors used there non-consecutive tuples, which outperforms consecutive tuples in some cases.

Our approach is, in a sense, close to the technique used in [7], with however significant differences: authors use alternating least square methods for optimization (in our case, all the matrices are real-valued and, softmax is applied whenever needed) and for the continuous case, they assume that the observation densities are a mixture of the predefined number of kernels.The "softmax trick" is also used in [9] (their model learns so-called *dense* representations of hidden states and observation probabilities – it is designed only for discrete HMMs).

As already mentioned, typically continuous HMMs assume that observations were sampled from a Gaussian distribution or a mixture of such distributions. For a non-Gaussian distribution, one usually

assumes some parametric family of distributions. In [11] authors propose a *decoupling* method to learning the parametric HMMs which are stationary. Instead of estimating the parameters of hidden states and observations jointly, they learn the parameters of observation densities using some parametric mixture learner, and then hidden states by solving some convex quadratic programming problem. In [13] a method is proposed for a case where at least one state-dependent distribution is modeled with some nonparametric technique (*e.g.*, maximum likelihood estimation under shape constraints), some broader review of methods is presented in [26]. In [14] authors use a Gaussian copula to model the dependence structure. A semiparametric data transformation is also proposed to ensure one may indeed use such copulas (the final observation distribution is a finite mixture of the copula models). In [27] authors present an efficient learning for parametric continuous HMMs sampled at finite irregular time instants (they incorporate some ideas from the theory of continuous-time Markov chains).

In [28], the authors consider the extension of a Kalman filter (which you can think of as a version of a HMM with continuous both, observations and hidden states), where the *pseudo-observations* (as authors call it) are transformed through a normalizing flow producing the actual observation.

The most similar model to our approach was introduced in Ghosh et al. [29] and further developed in [23], where the authors employ normalizing flows in observation densities and apply it to classification problems. However, they utilize the flow with coupling layers that cannot be directly applied to one-dimensional data. We use CNFs that can be easily used for any type of data, and are characterized by better quality for low-dimensional examples compared to coupling-based flows. Moreover, their approach relies on EM training, which makes it impractical for longer sequences of data. We propose two variants of training models: a) $\mathcal{F}^{\mathbf{Q}}$ model which relies on co-occurrences of discretized data and can successfully be applied on very long sequences, and b) $\mathcal{F}^{\mathrm{ML}}$ model which applies gradient-based optimization not only to the parameters of the flow (as in [29] and [23]), but also to the transition matrix on hidden states (which - because the underlying chain is stationary – is uniquely determined by the matrix representing joint states probabilities and parametrization given in Eqs. (8) and (9)) – consequently, in each iteration, only one forward step is needed, while Ghosh et al. model requires two forward passes (one to fit frozen weights (no gradient) and one gradient-based to update weights).

For one-dimensional examples we show the advantages of the proposed $\mathcal{F}^{\mathbf{Q}}$ empirically (compared to $\mathcal{F}^{\mathrm{ML}}$ model). Finally, the authors of [29] apply their model only on sequence classification as *black box*. At the same time, we investigate the proprieties of specific examples in order to discover the capabilities of distribution adjustment and discover the hidden states of data unobserved during training.

## 5 Experimental results

**Evaluation.** Here we consider our FlowHMM models $\mathcal{F}^{\mathrm{ML}}$ and $\mathcal{F}^{\mathbf{Q}}$ with training procedures described in Sections 3.1 and 3.2 respectively.

We compare our results with classic HMM models utilizing Gaussian mixtures observation densities with $K = 1, 10, 20$ components; we denote them by $\mathcal{G}^{(K)}$ and for simplicity we write $\mathcal{G} \equiv \mathcal{G}^{(1)}$. We used `hmmlearn` library[1] for the computations of all $\mathcal{G}^{(K)}$ models and provide the source code for FlowHMM models[2]. We evaluate the quality of the proposed methods computing *Normalized log-likelihood* (`normLL`) of unseen observations, *Total variation distance* ($d_{\mathrm{tv}}$) between learned and true emission probabilities, and accuracy of predicted hidden states (`accuracy`).

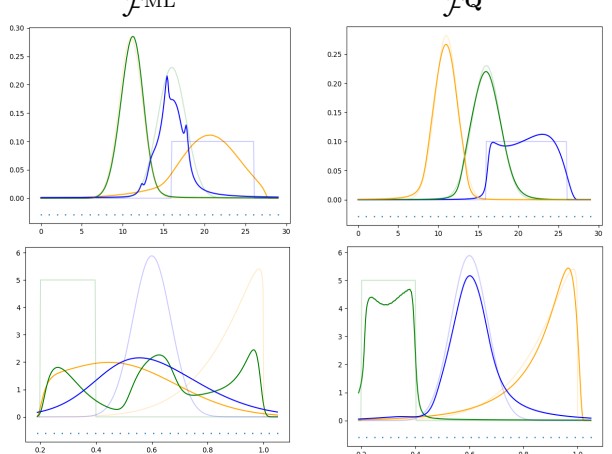

Figure 2: Distributions learned by $\mathcal{F}^{\mathrm{ML}}$ and $\mathcal{F}^{\mathbf{Q}}$ for $T = 10^4$ observations for Example 1 (top row) and Example 2a (bottom row). Original distributions shaded.

---

[1] `https://github.com/hmmlearn`
[2] `https://github.com/tooploox/flowhmm`

| | $T$ | Example 1 | | | | | Example 2a | | | | | Example 2b | | | | |
|---|---|---|---|---|---|---|---|---|---|---|---|---|---|---|---|---|
| | | $\mathcal{G}$ | $\mathcal{G}^{(10)}$ | $\mathcal{G}^{(20)}$ | $\mathcal{F}^{\mathrm{ML}}$ | $\mathcal{F}^{\mathbf{Q}}$ | $\mathcal{G}$ | $\mathcal{G}^{(10)}$ | $\mathcal{G}^{(20)}$ | $\mathcal{F}^{\mathrm{ML}}$ | $\mathcal{F}^{\mathbf{Q}}$ | $\mathcal{G}$ | $\mathcal{G}^{(10)}$ | $\mathcal{G}^{(20)}$ | $\mathcal{F}^{\mathrm{ML}}$ | $\mathcal{F}^{\mathbf{Q}}$ |
| normLL | $10^3$ | -2.21 | **-2.21** | -2.21 | -2.46 | -3.06 | 0.17 | 0.17 | 0.17 | **0.33** | 0.11 | 0.09 | 0.17 | 0.17 | **0.32** | 0.09 |
| | $10^4$ | -2.19 | -2.19 | -2.19 | -2.28 | **-2.18** | 0.18 | 0.17 | 0.17 | 0.33 | **0.34** | 0.09 | 0.18 | 0.18 | **0.32** | 0.30 |
| | $10^5$ | -2.19 | -2.19 | -2.19 | -2.28 | **-2.16** | 0.18 | 0.17 | 0.17 | **0.34** | 0.33 | 0.09 | 0.18 | 0.18 | **0.32** | 0.18 |
| $d_{\mathrm{tv}}$ | $10^3$ | **0.09** | 0.09 | 0.09 | 0.13 | 0.39 | **0.28** | 0.31 | 0.31 | 0.38 | 0.43 | – | – | – | – | – |
| | $10^4$ | 0.07 | 0.07 | 0.07 | 0.11 | **0.05** | 0.29 | 0.32 | 0.32 | 0.30 | **0.15** | – | – | – | – | – |
| | $10^5$ | 0.07 | 0.07 | 0.07 | 0.11 | **0.04** | 0.29 | 0.32 | 0.32 | 0.28 | **0.15** | – | – | – | – | – |

Table 1: `normLL` and $d_{\mathrm{tv}}$ metrics for synthetics Examples 1, 2a-b. Best results **bolded**.

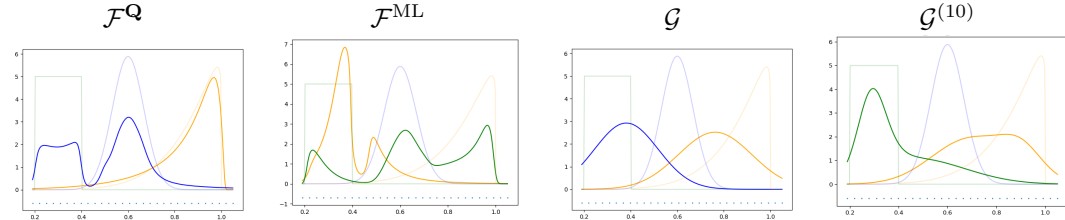

$\mathcal{F}^{\mathbf{Q}}$      $\mathcal{F}^{\mathrm{ML}}$      $\mathcal{G}$      $\mathcal{G}^{(10)}$

Figure 3: Results for Example 2b (three original distributions shaded) for $T = 10^4$ observations.

We define all the metrics and motivate their choice in A.2, the experimental settings are described in details in A.3.

**1D Synthetic sequences.** We consider HMMs with $L = 3$ hidden states and experiment with transition matrices $\mathbf{A}_1$ (used in [7, Eq. (11)]) or $\mathbf{A}_2$, where

$$\mathbf{A}_1 = \begin{pmatrix} 0.0 & 0.9 & 0.1 \\ 0.0 & 0.0 & 1.0 \\ 1.0 & 0.0 & 0.0 \end{pmatrix}, \quad \mathbf{A}_2 = \begin{pmatrix} 0.4 & 0.2 & 0.4 \\ 0.25 & 0.5 & 0.25 \\ 0.4 & 0.2 & 0.4 \end{pmatrix}. \tag{17}$$

We consider 2 examples. **Example 1** - transition matrix $\mathbf{A}_1$ and emission probabilities: two Gaussian and one Uniform. **Example 2a** - transition matrix $\mathbf{A}_2$, emission probabilities: one Gaussian, one Uniform, and Beta distribution. In addition, we consider **Example 2b**, where we aim at adjusting the models assuming only two hidden states. The examples are described in details in Appendix A.4.

For each example we sample train and test observations of the same length $T \in \{10^3, 10^4, 10^5\}$. We train $\mathcal{F}^{\mathrm{ML}}$ and $\mathcal{F}^{\mathbf{Q}}$ models using 500 (Example 1) and 1000 (Examples 2a and 2b) epochs. For $\mathcal{F}^{\mathbf{Q}}$ we used a grid of size $M = 30$. For each case (*i.e.*, fixed example and $T$) we perform 10 simulations and report means of `normLLs` and total variation distances in Table 1. For observations of length $T \geq 10^4$ flow models clearly outperform Gaussian ones in all the examples. Note that in these examples we had a grid of size $M = 30$, thus we estimated the matrix $\mathbf{Q}$ with 900 entries (number of observation pairs). It is intuitively clear that $T = 10^3$ observations is not enough then for $\mathcal{F}^{\mathbf{Q}}$. In this case $\mathcal{F}^{\mathrm{ML}}$ outperforms $\mathcal{F}^{\mathbf{Q}}$ in terms of `normLLs`; models are comparable for larger $T$. For $T \geq 10^4$ the $\mathcal{F}^{\mathbf{Q}}$ model "recovers" original distributions much more accurately, the small values of $d_{\mathrm{tv}}$ from Table 1 are confirmed in Fig. 2, where distributions learned by our flow models trained with $T = 10^4$ observations are depicted (from single simulations). In A.4 in Fig. 5, the trained distributions for Examples 1, 2a-b trained with $T = 10^3$ observations are depicted. In A.7 we investigate the impact of length $T$ on quality of the model.

We encourage the reader to confront Fig. 2 for $\mathcal{F}^{\mathbf{Q}}$ and $T = 10^4$ for Example 1 with [7, Fig. 4]. In Fig. 3 we depict the distributions learned models with only $L = 2$ hidden states for $T = 10^4$ observations (recall observations were sampled from a model with 3 different distributions). Since flow models are capable of fitting multimodal distributions, they outperform Gaussian ones. Moreover, the co-occurrence matrix-based model $\mathcal{F}^{\mathbf{Q}}$ outperforms the maximumum-likelihood-based model $\mathcal{F}^{\mathrm{ML}}$. Recall that, by construction, our models $\mathcal{F}^{\mathrm{ML}}$ and $\mathcal{F}^{\mathbf{Q}}$ assume that the underlying Markov chain is stationary. We want to point out that in our examples we used at most $L = 3$ states. In such a case, even if the assumption of stationarity does not hold, it does not have a large impact on results. This is because for such chains the rate of convergence to stationarity is exponential (in number of steps). To be more specific, *e.g.*, for the chain with t.m. $\mathbf{A}_2$ given in (17), after 10 steps the total variation between the stationary distribution (which is $\pi(1) = 5/14, \pi(2) = 4/14, \pi(3) = 5/14$) and the distribution of $X_{10}$ is at most (for any initial distribution) $4.22 \cdot 10^{-6}$.

**Real datasets. Example 3.** S&P 500 and Dow Jones are popular measures of market performance. We retrieved data [30] for a period of 9/1977-8/2017 that consist of 2082 points. We trained models on $T = 1000$ observations (and computed `normLLs` for the remaining data).

**Example 4.** We used mean air pressure from [31] dataset collected between 02/15/18 and 02/28/19. There were 54 531 observations, the models were trained on the first $T = 40\,000$ ones and tested them on the remaining observations. In both, Example 3 and Example 4, we applied a standard difference transform (to remove a trend), *i.e.*,

|   | S&P 500 | | | | |
|---|---|---|---|---|---|
| $L$ | $\mathcal{G}$ | $\mathcal{G}^{(10)}$ | $\mathcal{G}^{(20)}$ | $\mathcal{F}^{\mathrm{ML}}$ | $\mathcal{F}^{\mathbf{Q}}$ |
| 2 | -10.476 | -10.157 | -10.686 | -6.850 | **-5.479** |
| 3 | -10.800 | -10.194 | -10.806 | -6.837 | **-4.966** |
| 4 | -12.462 | -9.7641 | -11.130 | -6.908 | **-5.043** |
| | **Dow Jones** | | | | |
| 2 | -12.317 | -14.154 | -13.138 | -8.730 | **-7.120** |
| 3 | -12.645 | -13.747 | -15.007 | -8.897 | **-7.426** |
| 4 | -13.956 | -13.313 | -14.669 | -8.902 | **-7.514** |
| | **Air pressure** | | | | |
| 2 | -0.704 | 0.169 | 0.598 | -0.984 | **0.823** |
| 3 | -0.743 | 0.646 | 0.603 | 0.540 | **0.861** |
| 4 | -0.758 | 0.710 | 0.692 | 0.661 | **0.783** |

Table 2: `normLL` for Examples 3 and 4.

we considered time series $y'_k = y_k - y_{k-1}$. We trained the models with $L \in \{2, 3, 4\}$ hidden states, models $\mathcal{F}^{\mathrm{ML}}$ and $\mathcal{F}^{\mathbf{Q}}$ were trained for 500 epochs, for the latter we used a uniform grid of size $M = 30$. During training $\mathcal{F}^{\mathrm{ML}}$ model we randomly sampled subsequences of length $10^3$ for each epoch. The values of `normLL` are reported in Table 2. As can be seen, in all cases $\mathcal{F}^{\mathbf{Q}}$ yields the best results and for Example 3 model $\mathcal{F}^{\mathrm{ML}}$ is second best. Note also that although a number of observations for Example 3 is not large (we used only $T = 1000$ observations, while for $M = 30$ we have 900 observation pairs) $\mathcal{F}^{\mathbf{Q}}$ outperforms $\mathcal{F}^{\mathrm{ML}}$. Fig. 6 in A.4 shows that the learned observation densities for $L \in \{2, 3, 4\}$ are non-Gaussian. We want to remark one thing: in financial data one usually works with log-returns $\log(y_k/y_{k-1})$, which tend to be "*more Gaussian*". This is also the case with the examples we considered: $\mathcal{F}^{\mathrm{ML}}, \mathcal{F}^{\mathbf{Q}}$ and Gaussian models returned very similar `normLLs` (which is no surprise in case of Gaussian data, other models easily fitted to them). We chose and reported the differences $y_k - y_{k-1}$ to compare the models in case of non-Gaussian data.

**2D synthetic sequences.** We consider HMM models with $L = 3$ states. We consider 2 examples: **Example 5** with emission probabilities: two "`Moons`" and one Uniform and **Example 6** with emission probabilities: one bivariate Gaussian, one Uniform and one related to geometric Brownian motion. Details provided in A.5. In both examples we consider the transition matrix $\mathbf{A}_1$ (variant **(a)**) and $\mathbf{A}_2$ (variant **(b)**), defined in (17).

| | $T$ | | | Example 5a | | | | | | | Example 5b | | | |
|---|---|---|---|---|---|---|---|---|---|---|---|---|---|---|
| | | $\mathcal{G}$ | $\mathcal{G}^{(10)}$ | $\mathcal{G}^{(20)}$ | Ghosh | $\mathcal{F}^{\mathrm{ML}}$ | $\mathcal{F}^{\mathbf{Q}}$ | $\mathcal{G}$ | $\mathcal{G}^{(10)}$ | $\mathcal{G}^{(20)}$ | Ghosh | $\mathcal{F}^{\mathrm{ML}}$ | $\mathcal{F}^{\mathbf{Q}}$ |
| normLL | $10^3$ | -1.6572 | -0.6521 | -0.6637 | -0.7643 | **-0.6003** | -2.7090 | -1.6560 | -1.4459 | -1.4486 | **-0.7430** | -1.3309 | -2.6789 |
| | $10^4$ | -1.6083 | -0.6176 | -0.6081 | -0.7052 | **-0.4822** | -0.6711 | -1.6574 | -1.4392 | -1.4356 | **-0.7036** | -1.2286 | -1.4185 |
| | $10^5$ | -1.6117 | -0.6001 | -0.5988 | -0.6740 | **-0.4898** | -3.4500 | -1.6510 | -1.4160 | -1.4178 | **-0.6728** | -1.2538 | -3.0717 |
| accuracy | $10^3$ | 0.6010 | **1.0000** | 1.0000 | – | 1.0000 | 0.5600 | 0.5450 | 0.6560 | 0.6480 | – | **0.7390** | 0.5360 |
| | $10^4$ | 0.6325 | 0.9995 | **0.9998** | – | 0.9998 | 0.9956 | 0.5680 | 0.6677 | 0.6723 | – | 0.7088 | **0.7414** |
| | $10^5$ | 0.6274 | **1.0000** | 1.0000 | – | 0.9998 | 0.6097 | 0.5726 | 0.6775 | 0.6727 | – | **0.7448** | 0.5615 |
| | | | | Example 6a | | | | | | | Example 6b | | | |
| normLL | $10^3$ | 0.8652 | 0.9443 | 0.9472 | 0.1600 | **1.0531** | -2.1823 | 0.2270 | 0.3128 | 0.2915 | 0.1487 | **0.3636** | -2.1891 |
| | $10^4$ | 0.8755 | 1.0009 | 1.0014 | 0.1673 | **1.0807** | 0.9189 | 0.2378 | 0.3161 | 0.3173 | 0.1665 | **0.3551** | 0.2120 |
| | $10^5$ | 0.8843 | 0.9978 | 0.9992 | 0.1713 | **1.0688** | -1.8349 | 0.2186 | 0.3076 | 0.3103 | 0.1730 | **0.3496** | -0.3917 |
| accuracy | $10^3$ | **0.9960** | 0.9960 | 0.9960 | – | 0.9940 | 0.6700 | 0.8270 | 0.6580 | 0.6590 | – | **0.8550** | 0.6490 |
| | $10^4$ | 0.9971 | 0.9987 | 0.9987 | – | **0.9989** | 0.9934 | 0.8173 | 0.6533 | 0.6558 | – | 0.5419 | **0.8428** |
| | $10^5$ | 0.9964 | **0.9994** | 0.9994 | – | 0.9987 | 0.9219 | 0.8077 | 0.6503 | 0.6443 | – | **0.8796** | 0.7376 |

Table 3: `normLL` and `accuracy` for Examples 5 and 6. Best results **bolded**.

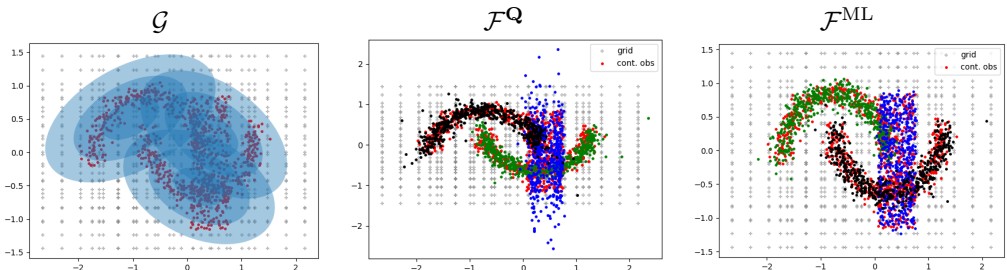

Figure 4: Trained distributions for Example 5a for $T = 10^4$: original observations denoted as red dots •, grid $\Gamma$ presented as gray crosses $+$; samples from trained distributions $f_{\boldsymbol{\beta}_l}, l = 1, 2, 3$ depicted as black •, green • and blue • dots (colors correspond to state $l$).

We compare Gaussian baseline models, our $\mathcal{F}^{\text{ML}}$ and $\mathcal{F}^{\mathbf{Q}}$ models, as well as a model from Ghosh at al. [23] – we used authors' official implementaion[3]. For the latter we computed only `normLL`s.

In Table 3 values of `normLL` (for all models) and `accuracy` (for all models except Ghosh at al. [23]) are reported, whereas in Fig. 4 the trained distributions for Example 5a for $T = 10^4$ are depicted. Similar plots for Example 6a are presented in A.5 in Fig. 7. We used non-uniform grids of size $M = 30^2$ (cartesian product of two one-dimensional grids of size 30 on each coordinate – for other strategies for choosing a grid see A.1. One can observe that for all examples except Example 5b model $\mathcal{F}^{\text{ML}}$ gives the best `normLL`s, Ghosh et al. outperforms our models only in Example 5b. For 2D examples we can spot a drop of quality – in terms of `normLL` – for $\mathcal{F}^{\mathbf{Q}}$ model. We performed additional simulations for this model for $T = 10^3$ observations and grids of sizes $35^2$ and $40^2$, the results are provided in Table 4.

As we can see they are still worse than $\mathcal{F}^{\text{ML}}$ model (which is no surprise: we have $30^4, 35^4$ and $40^4$ pairs and only $10^3$), but we see a tendency: increasing a grid size increases `normLL` in all cases. One can observe that in all the considered cases and variants, model $\mathcal{F}^{\text{ML}}$ gives the best `normLL` values. When it comes to the other metric, *i.e.*, the `accuracy`, all the classic Gaussian mixture models, as well as $\mathcal{F}^{\text{ML}}$, yield almost the same accuracy if transition matrix $\mathbf{A}_1$ was used (variant (**a**)). However, for the transition matrix $\mathbf{A}_2$ (variant (**b**)), the models $\mathcal{F}^{\text{ML}}$ and $\mathcal{F}^{\mathbf{Q}}$ outperform the classic models with significant margin (note that $\mathbf{A}_2$ yields a "more complex" model compared to $\mathbf{A}_1$).

| Ex. | $M = 35^2$ | $M = 40^2$ |
|---|---|---|
| 5a | -2.548 | -2.499 |
| 5b | -2.520 | -2.470 |
| 6a | -2.042 | -1.997 |
| 6b | -2.042 | -1.990 |

Table 4: Values of `normLL` for $\mathcal{F}^{\mathbf{Q}}$ model for Examples 5a-b, 6a-b for $T = 10^3$ observations.

**High dimensional (6D) real dataset.** **Example 7**: from [31] dataset (used in Example 4) we chose 6 features related to humidity (`mean, stdev`) and air pressure (`min, max, mean, stdev`). Similarly as in Example 4 we considered differences of the observations. Again, the length of training set was set to $40k$, the remaining $\sim 14k$ observations constituted a test set. We used $\mathcal{F}^{\text{ML}}$ model – in each epoch we randomly sampled a subsequence of length $10^3$. In all cases ($L = 2, 3, 4$) the model significantly outperfoms Gaussian models, as can be seen in Table 5.

| $L$ | $\mathcal{G}$ | $\mathcal{G}^{(10)}$ | $\mathcal{G}^{(20)}$ | $\mathcal{F}^{\text{ML}}$ |
|---|---|---|---|---|
| 2 | -1097.49 | -1096.81 | -1096.65 | **-571.08** |
| 3 | -1097.45 | -1096.73 | -1096.51 | **-571.97** |
| 4 | -1097.39 | -1096.65 | -1096.09 | **-572.61** |

Table 5: Values of `normLL` for 6-dimensional data.

## 6 Conclusions and limitations

In this work, we proposed a continuous hidden Markov model that leverages deep flow-based network architectures to model complex, non-Gaussian distributions. Although our approach outperforms several baselines and competing approaches, it relies on the co-occurrence matrix that can be only computed with the assumption that the Markov chain on a set of hidden states is stationary - a limitation not present in the EM-based approaches. Addressing this constraint, *e.g.*, by re-computing temporally stationary co-occurrence matrix during training, is part of our future work. So is in-depth validating our model on multivariate time series or distributions, where our flexible approach can potentially offer more benefits over the existing works. While we do not identify any straightforward negative societal implications of our work, we acknowledge that its application to large data corpora can lead to significant energy consumption, despite our main contributions being focused on increased efficiency.

## Acknowledgements

This work was supported by Foundation for Polish Science (grant no POIR.04.04.00-00-14DE/18-00) carried out within the Team-Net program co-financed by the European Union under the European Regional Development Fund, as well as the National Centre of Science (Poland) Grant No. 2020/39/B/ST6/01511. The work conducted by Maciej Zieba was supported by the National Centre of Science (Poland) Grant No. 2021/43/B/ST6/02853. For the purpose of Open Access, the author has applied a CC-BY public copyright licence to any Author Accepted Manuscript (AAM) version arising from this submission.

---

[3]`https://github.com/anubhabghosh/genhmm`

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
