# A Appendix

## A.1 Continuous observations and discretization.

For the purpose of the training process of $\mathcal{F}^{\mathbf{Q}}$, the continuous sequence of observations must be discretized first. Assume our observations are from $\mathbb{R}^d, d \geq 1$. Let $\Gamma = (\gamma_1, \dots, \gamma_M), \gamma_i \in \mathbb{R}^d$, be some *grid* of size $M$. The grid may be predefined, created uniformly, or may represent the centroids of clustering approach (of the training set). Having continuous observations $y_{0:T} = (y_0, \dots, y_T)$ we round each $y_i$ to the nearest point from $\Gamma$ obtaining a discretized sequence $y_{0:T}^{\Gamma} = (y_0^{\Gamma}, \dots, y_T^{\Gamma})$, where:

$$y_i^{\Gamma} = \arg\min_{\gamma \in \Gamma} ||\gamma - y_i||^2, \quad i = 0, 1, \dots, T. \tag{18}$$

The values of the grid represent the set of discrete observations $\mathcal{V}$, *i.e.*, $\mathcal{V} = \Gamma$. The simplest uniform grid would be following: take $\min$ and $\max$ of each coordinate of observations and for each dimension construct a uniform grid of size $m$. Then as the final grid we may take a Cartesian product of this one-dimensional grids. The resulting grid would be of size $M = m^d$ and the co-occurence matrix $\hat{\mathbf{Q}}$ would have $m^{2d}$ entries. On one hand, $m$ cannot be too small, since the discretised observations should be fairly close to original ones. On the other hand, $m$ cannot be too large, since the size of $\hat{\mathbf{Q}}$ would be prohibitively large then. That is why we use the uniform grid for our univariate examples (Example 1 and 2a-b), whereas we propose (and use for two-dimensional Examples 5 and 6) the following strategy.

**Strategy for grid choice**. Fix some relatively small $m$ and $\rho \in (0, 1)$. For simplicity assume that $m_0 = \rho m$ is an integer. Let $y_{0:T}^{\text{train}}(i)$ denote a vector of $i$-th feature of observations, from which we construct a one-dimensional grid $\Gamma^{i,a} = (\gamma_1^{i,a}, \dots, \gamma_{\rho m_0}^{i,a})$, taking $\gamma_1^{i,a} = \min(y_{0:T}^{\text{train}}(i))$ and $\gamma_{\rho m}^{i,a} = \max(y_{0:T}^{\text{train}}(i))$. Additionally, construct a grid $\Gamma^{i,b}$ in the following way: perform $k$-means on $y_{0:T}^{\text{train}}(i)$ with $k = (1 - \rho)m$ and take the centers of the clusters as the grid. Then we set $\Gamma^i = \Gamma^{i,a} \cup \Gamma^{i,b}$. As the final grid we take the Cartesian product

$$\Gamma = \Gamma^1 \times \dots \Gamma^d.$$

In this strategy we still have $M = |\Gamma| = m^d$, however $m$ can be significantly smaller than in the uniform case. We always assure that there are some *relevant* points (close to continuous observations) and some *far away* points (we need *e.g.*, pairs of *relevant* and *far away* points for proper training).

## A.2 Metrics used for evaluation

*Normalized log-likelihood.* In all the examples, flow model $\mathcal{F}^{\mathbf{Q}}$ is trained on $y_{0:T}^{\text{train}, \Gamma}$, whereas $\mathcal{F}^{\text{ML}}$ and Gaussian models are trained on $y_{0:T}^{\text{train}}$. For each model we compute `normLL` – the normalized log-likelihood of continuous observations $y_{0:T'}^{\text{test}}$, *i.e.*, $\log p(y_{0:T'}^{\text{test}}; \boldsymbol{\theta})/(T' + 1)$ (see Eq. (2)). To compute it for flow models we used the standard forward algorithm.

Moreover, for univariate synthetic examples (for which we know the observations densities $p(y|s_i)$) we also determine how well the model learned the observation densities and how well it inferred the hidden states (since we know the actual hidden states at each step).

*Total variation distance.* Note that we may evaluate $f_{\boldsymbol{\beta}_i}(y)$ at finite number of points $y \in \mathbb{R}$, we must thus do it on some grid. We will use another, more dense, uniform grid $\Gamma' = (\gamma_1', \dots, \gamma_r')$, where $\gamma_1' = \gamma_1, \gamma_r' = \gamma_M$. We take $r = 10M$ in all the examples. For two continuous distributions $g_1$ and $g_2$ we compute the total variation distance as

$$d_{\text{tv}}(g_1, g_2) = \frac{1}{2} \sum_{k=1}^{r} |g_1(\gamma_k') - g_2'(\gamma_k')|$$

(where we actually normalize $g_1$ and $g_2$ so that they are distributions on $\Gamma'$). To compare distributions $p(\cdot|s_i)$ with with $f_{\boldsymbol{\beta}_i}(\cdot), i = 1, \dots, L$ (*i.e.*, with distribution learned by flow models), we compute the mean of total variation distances

$$d_{\text{tv}} = \frac{1}{L} \sum_{i=1}^{L} d_{\text{tv}}(p(\cdot|s_i), f_{\boldsymbol{\beta}_i}(\cdot)) = \frac{1}{2L} \sum_{i=1}^{L} \sum_{k=1}^{r} |p(\gamma_k'|s_i) - f_{\boldsymbol{\beta}_i}(\gamma_k')|.$$

Similarly, for Gaussian models we compute

$$d_{\text{tv}} = \frac{1}{L} \sum_{i=1}^{L} d_{\text{tv}} \left( p(\cdot|s_i), \sum_{k=1}^{K} \alpha_{i,k} \mathcal{N}\left(\cdot; \mu_{i,k}, (\sigma_{i,k})^2\right) \right),$$

where $\{\alpha_{i,k}, \mu_{i,k}, (\sigma_{i,k})^2\}_{k=1}^{K}, i = 1, \ldots, L$ are the parameters learned by $\mathcal{G}^K$.

*Accuracy of predicted hidden states.* For synthetic examples we know the true hidden states $x_{0:T}^{\text{test}}$. We implemented the Viterbi algorithm to compute predicted hidden states $x_{0:T}^{\text{pred}}$ for our $\mathcal{F}^{\mathbf{Q}}$ and $\mathcal{F}^{\text{ML}}$ models, then we simply compute the `accuracy` between $x_{0:T'}^{\text{test}}$ and $x_{0:T'}^{\text{pred}}$.

### A.3 Experimental settings

We use ADAM [32] as an optimizer with $lr = 0.01$, default betas $(0.9, 0.999)$, and weight decay equal $0.0001$. In all experiments, we use two blocks of CNFs with *concatsquash* layers, batch norm, and the configuration of the hidden neurons equal $16 - 16$.

### A.4 Examples 1D - details

We sample the observations from the following distributions: uniform distribution $\mathcal{U}(a, b)$, normal distribution $\mathcal{N}(\mu, \sigma^2)$, and beta distribution $\text{Beta}(\alpha, \beta)$ with shape parameters $\alpha > 0, \beta > 0$. The latter has the density function $p(y) \propto y^{\alpha-1}(1-y)^{\beta-1}$ for $y \in (0, 1)$.

Example 1: we used

$$p(\cdot|s_1) \sim \mathcal{N}(11, 2), \quad p(\cdot|s_2) \sim \mathcal{N}(16, 3) \quad \text{and} \quad p(\cdot|s_3) \sim \mathcal{U}(16, 26),$$

Example 2: we used

$$p(\cdot|s_1) \sim \text{Beta}(7, 1.1), \quad p(\cdot|s_2) \sim \mathcal{U}(0.2, 0.4) \quad \text{and} \quad p(\cdot|s_3) \sim \mathcal{N}(0.6, 0.0046).$$

To be consistent with discretization in [7], in Example 1, we take the integer grid points $\gamma = \{0, 1 \ldots, M - 1\}$, while in Example 2, we consider uniform grid $\Gamma = (\gamma_1, \ldots, \gamma_M)$ with $\gamma_1 = \min(y_{0:T}^{\text{train}})$ and $\gamma_M = \max(y_{0:T}^{\text{train}})$; we set the grid size $M = 30$.

In Fig. 5 we present the trained distributions for Examples 1, 2a-b for models trained on $T = 10^3$ observations. It is worth noting that the distributions learned by $\mathcal{G}^{10}$ are similar to those learned by $\mathcal{G}$. More precisely, each mixture of $\mathcal{G}^{10}$ is, roughly speaking, a mixture of 10 almost identical normal distributions with almost equal weights. Recall, in Fig. 2 the distributions learned for Examples 1 and 2a for $T = 10^3$ were presented.

In Fig. 6 the distributions trained by $\mathcal{F}^{\mathbf{Q}}$ for Examples 3 and 4 are depicted.

### A.5 Examples 2D

We consider HMM models with $L = 3$ hidden states and the observations sampled in the following ways:

Example 5:

- $p(\cdot|s_1)$ and $p(\cdot|s_2)$ are sampled using `make_moons` (which we call "*Moons*") from `sklearn.datasets` library,
- $p(\cdot|s_3)$ is a uniform distribution on $[0.8, 1.5] \times [-1, 1]$.

Example 6:

- $p(\cdot|s_1) \sim \mathcal{N}(\boldsymbol{\mu}, \boldsymbol{\Sigma})$ with

$$\boldsymbol{\mu} = (0.5, 0.5)^T, \quad \boldsymbol{\Sigma} = \begin{pmatrix} 0.1 & -0.014 \\ -0.014 & 0.04 \end{pmatrix},$$

- $p(\cdot|s_2)$ is a uniform distribution on $[0.25, 0.5] \times [0.7, 2.0]$,

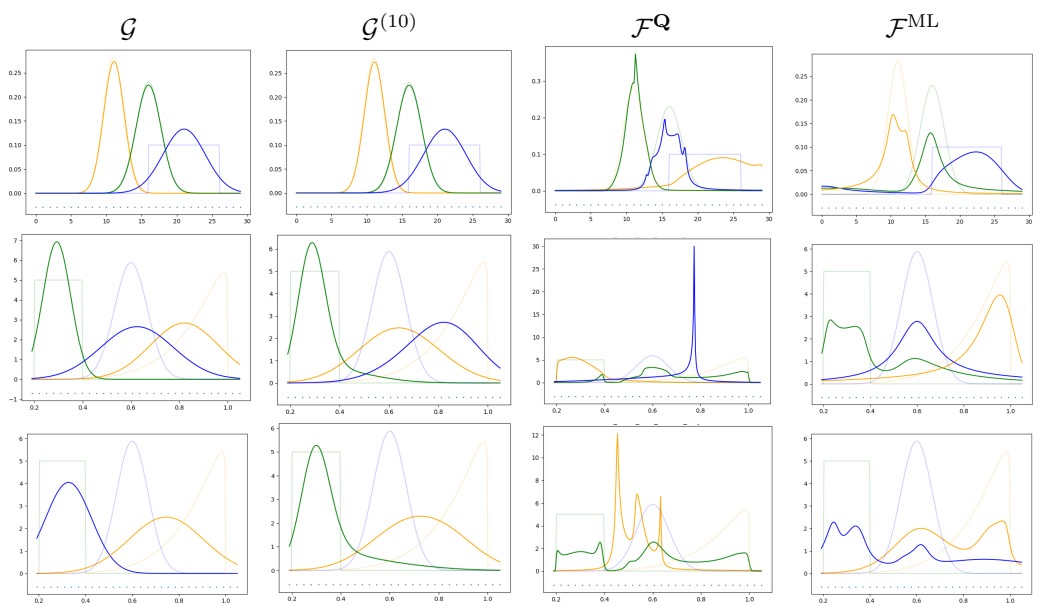

Figure 5: Trained distributions (three original distributions shaded) for $T = 10^3$ observations for Example 1 (first row), Example 2a (middle row) and Example 2b (bottom row).

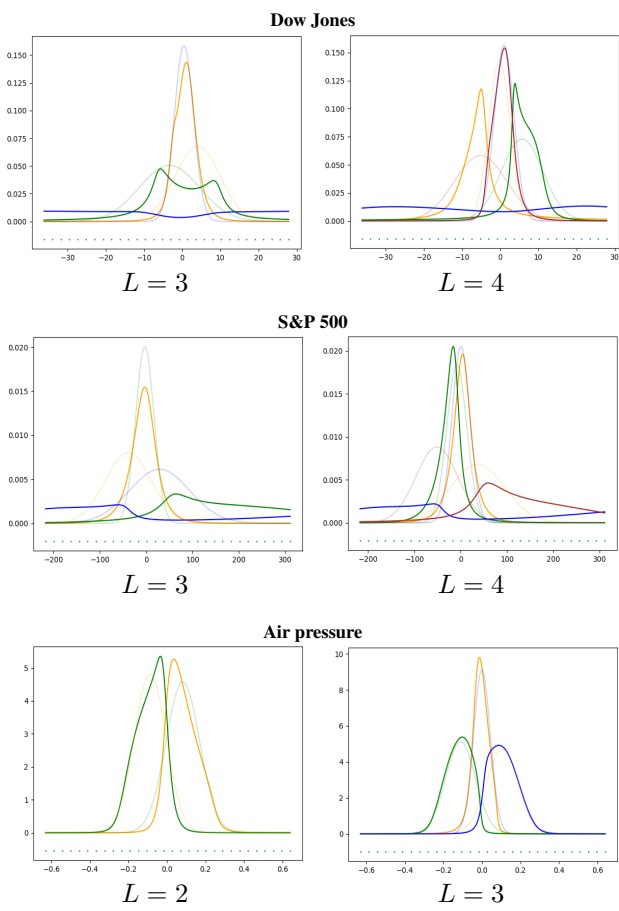

Figure 6: Distributions learned by $\mathcal{F}^{\mathbf{Q}}$ (colored) with $L \in \{2, 3, 4\}$ hidden states and $\mathcal{G}$ (grayed-out) for Examples 3 and 4 (Dow Jones, S&P 500 and Air pressure datasets), datasets (models with $L \in \{3, 4\}$ hidden states) .

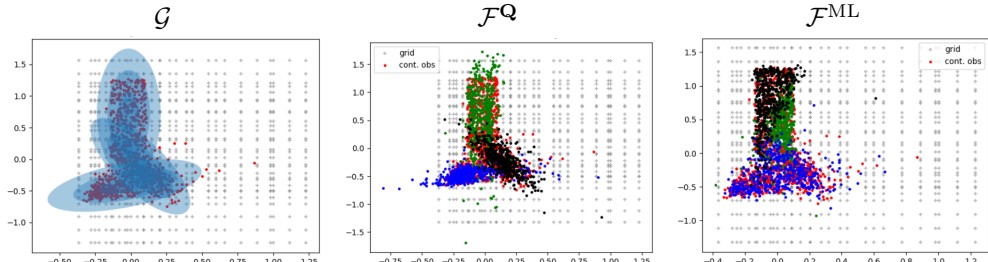

Figure 7: Trained distributions for Example 6a for $T = 10^4$: original observations denoted as red dots •, grid $\Gamma$ presented as gray crosses +; samples from trained distributions $f_{\boldsymbol{beta}_l}, l = 1, 2, 3$ depicted as black •, green • and blue • dots (colors correspond to state $l$). The similar plots for Example 5a were presented in Fig. 4.

- $p(\cdot|s_3)$ is a density of $(S(0.5), S(1))$, where $S(t)$ is a geometric Brownian motion, *i.e.*, $S(t) = S(0) \exp(\mu^* t + \sigma B(t))$, and $B(t)$ is a standard Brownian motion; we used the parameters: $S(0) = 0.3$, volatility $\sigma = 0.5$ and interest rate $r = 0.05$ ($\mu^* = r - \sigma^2/2$).

All observations were centered – centers were computed from the training set and then they were subtracted from both, the training and test set. In both examples we used *strategy for grid choice* described in A.1 with $m = 30$ and $\rho = 0.5$. Thus, the grid was of size $M = 30^2 = 900$.

In Fig. 7 the trained distributions for Example 6a for models trained with $T = 10^4$ observations are depicted. Recall, similar plots for Example 5a were presented in Fig. 4.

## A.6 Training time evaluation

We analyse the training time, by comparing the time/per epoch in sec. between $\mathcal{F}^{\mathbf{Q}}$, and $\mathcal{F}^{\mathrm{ML}}$. The experiment was performed on *NVIDIA GeForce RTX 2070 8GB*. The results of the test are provided in Table 6. The $\mathcal{F}^{\mathrm{ML}}$ can be efficiently trained for $T \leq 10^3$, therefore, for $T > 10^3$ we randomly sample subsequences of a given size $T = 10^3$ at each step.

| T | $\mathcal{F}^{\mathbf{Q}}$ | $\mathcal{F}^{\mathrm{ML}}$ |
|---|---|---|
| $10^1$ | 0.366 ($\pm$0.031) | 0.456 ($\pm$0.038) |
| $10^2$ | 0.414 ($\pm$0.015) | 0.529 ($\pm$0.041) |
| $10^3$ | 0.436 ($\pm$0.034) | 1.070 ($\pm$0.021) |
| $10^4$ | 0.492 ($\pm$0.034) | 6.236 ($\pm$0.097) |
| $10^5$ | 0.563 ($\pm$0.062) | 59.378 ($\pm$1.611) |

Table 6: The average training time per epoch (in sec.), for different lengths of training sequence.

## A.7 Impact of $T$ length

To investigate the influence of length of a sampled subsequence in each epoch of $\mathcal{F}^{\mathrm{ML}}$ training, we additionally tested what happens in cases of lengths $400, 2k, 4k, 10k, 40k$ (note, the latter actually means that all observations were used in each epoch). The results for 500 epochs are reported in Table 8. As we can see, the case of $T' = 2k$ is the best in case $L = 3, 4$ and $T' = 40k$ in case $L = 2$. It is worth noting that in this case $\mathcal{F}^{\mathrm{ML}}$ normLLs are worse than those of $\mathcal{F}^{\mathbf{Q}}$. This is even the case if we perform more epochs: for $L = 2$ and training $\mathcal{F}^{\mathrm{ML}}$ for 2000 epochs and $T' = 40k$ we obtained normLL=0.756, whereas, as reported in Table 2, $\mathcal{F}^{\mathbf{Q}}$ yielded normLL=0.823. At the same time, training $\mathcal{F}^{\mathrm{ML}}$ took 16 hrs 22 min, whereas training $\mathcal{F}^{\mathbf{Q}}$ took 4 m 49 s (the differences for $L = 3$ or $L = 4$ are of similar order).

To investigate if $\mathcal{F}^{\mathrm{ML}}$ is more competitive with longer samples sequences and fewer epochs, we additionally trained model for 100, 200, 300 and 500 epochs and $T' = 40k$. The resulting normLLs and execution times are reported in Table 7.

| $T' =$ | | 100 | $40k$ | $40k$ | $40k$ | $40k$ |
|---|---|---|---|---|---|---|
| Epochs: | | 500 | 100 | 200 | 300 | 500 |
| $L = 2$ | normLL | -0.984 | **-0.086** | -0.1186 | -0.1615 | -0.256 |
| | time | 4m | 37m | 1h 47m | 2h 32m | 3h 4m |
| $L = 3$ | normLL | 0.541 | -0.011 | **0.653** | 0.502 | 0.528 |
| | time | 13m | 50m | 1h 17m | 2h 35m | 4h 53m |
| $L = 4$ | normLL | 0.661 | -0.038 | **0.801** | 0.725 | 0.7075 |
| | time | 14m | 36m | 1h 18m | 1h 58m | 4h 23m |

Table 7: **Air pressure** dataset: Competitiveness of $\mathcal{F}^{\mathrm{ML}}$ model with $T' = 40k$ and fewer epochs.

| $T' =$ | 500 | $1k$ | $2k$ | $4k$ | $10k$ | $40k$ |
|---|---|---|---|---|---|---|
| $L = 2$ | -0.631 | -0.984 | -0.572 | -0.568 | -0.725 | **-0.256** |
| $L = 3$ | 0.541 | 0.541 | **0.627** | 0.473 | 0.408 | 0.528 |
| $L = 4$ | 0.743 | 0.661 | **0.751** | 0.734 | 0.515 | 0.707 |

Table 8: **Air pressure** dataset: values of normLL for $\mathcal{F}^{\mathrm{ML}}$ trained with sampling a subsequence of length $T'$ in each epoch.

In a sense, training for fewer epochs and on whole observation sequence is competitive: *e.g.*, for 200 epochs normLLs are better, the price is longer execution time: for $L = 3$ or $L = 4$ it is c.a. 5 times larger, whereas for $L = 2$ it is c.a. 26 times larger.

## B  Broader impact

HMM models are still applied on many domains in research and industry, they achieve competitive results compared to heavy deep learning models. They may have very positive impact in drug discovery, or various medical applications. On the other hand they can be used in some inappropriate ways, for instance in surveillance tools. Moreover, we propose effective method of training our model, that reduces the computational costs for larger sequences.