# OpenReview forum: "FlowHMM: Flow-based continuous hidden Markov models"
_NeurIPS.cc/2022/Conference — NeurIPS 2022 Accept_

### Official Review · Reviewer_idya · 2022-07-03

**Rating:** 7
**Confidence:** 4
**Soundness:** 2 fair
**Presentation:** 3 good
**Contribution:** 2 fair

**Summary:**

The authors study the problem of fitting Hidden Markov Models (HMM) when the continuous observation model is non-Gaussian. Precisely, they consider a family (one for each state of the latent variable) of continuous normalising flow models as the observation model. To jointly fit the transition probabilities of the latent states as well as the continuous flow models, the authors propose two methodologies:
* FlowHMM \mathcal{F}^{EM} : maximising the log-likelihood, whilst parameterising the joint latent distribution through a soft-max.
* FlowHMM \mathcal{F}^{Q} : fit the observation co-occurrence matrix, whilst:
    - parameterising the emission probabilities through a soft-max of the normalizing flow
    - discretising the continuous observation, allowing the observation co-occurrence matrix to be substituted by the empirical variant.

The second method being motivated by the fact that directly optimising the log-likelihood is more computationally costly as the length of the time series increases.
The authors evaluate these two methods against Gaussian mixture model methods on both: 1 and 2 dimensional synthetic examples, as well as real world financial and air pressure datasets.

Post Rebuttal Edit: Thank you to the authors for providing follow up experiments and discussion. I have updated my score.

**Questions:**

Questions:
* **Computational Cost**: Following the weakness associated to the computational cost:
	- What is the impact of sub-sampling sequence length on the statistical performance of the EM method? Particularly, the air pressure data set where there are 40k time steps and only 1k is sampled at any given time.
	- Would the EM method be more competitive with longer sampled sequences and fewer epochs?

* For the real datasets, S&P 500 and Dow Jones, are the datasets prices? If so:
    - it is standard practice in financial applications to work with log returns \log(y_k/ y_{k-1}), which tend to be more Gaussian (albeit heavy tailed).
     - how do the Gaussian mixture models perform compare to the normalizing flow methods in this case?

* When Gaussian noise is added to the observations in equation (16)
	- how is the Gaussian noise hyperparameter chosen? (does it depend on the grid resolution ?)
	- what is its impact on training performance ?

Notations:

* Within background, the subscript notation for the parameters in x i.e. \alpha_{x,k}, \mu_{x,k}, \sigma^2_{x,k}, is overloaded when using \beta, as \beta_l = \alpha_{\ell,k}, \mu_{\ell,k}, \sigma^2_{\ell,k}. This should be made clearer.
* Equation (4) z_{n-1} needs to be defined
* Section 3.3 : "due ti the fact" ti -> to



**Limitations:**

* **Curse of dimensionality**: The method used to discretize the observations for higher-dimensional observations seems ad-hoc, with the experiments suggesting the EM method is better when the observation model is more than 2 dimensional.

**Strengths And Weaknesses:**

Strengths
* Extending HMM to non-Gaussian observation domains is likely an important problem, where normalizing flows potentially offer a natural and flexible solution.
* The manuscript is overall clearly written, with good introductions to the background and problem setting.

Weaknesses
* **Curse of dimensionality**: The approach of discretising the response tends to be limited to one (or low) dimensional observations for the co-occurrence method.
* **Computational cost**: Experiments provide few details on how the two proposed methods are fairly compared from a computational cost perspective. That is, the Appendix states (where T is the time length of the sequence) that "... for T > 1000 we randomly [for the EM model] sample sub-sequences of a given size T = 1000 at each step". The impact of this decision is neither discussed or investigated within the main body.

---

> ### Author Response · Authors · 2022-08-02
> **Official response to reviewer idya (part 1 of 2)**
>
> We are delighted to have the pleasure of taking up your suggestions upon our submission. We would like to clarify reviewer's open points which can bring our work closer to satisfy your high standards of research. We will be happy to respond to any of the Reviewer's further questions.
>
> Concerning **Computational Cost**:
>
> We chose subsampling $1k$ consecutive observations in each step of $\mathcal{F}^{\rm{EM}}$ training to reduce the execution time (there is a tradeoff between execution time [length of subsampled sequence] vs performance). If we allow “pay” more time for execution and sample subsequences of length $2k$ we would obtain even better results. To be more exact: we have performed exhaustive additional simulations for “Air pressure” (Example 4) dataset, which we are willing to report in a revised version. We tested several  versions of sampling subsequences’ lengths, namely, in addition to $10^3$, we considered   500, 2000, 4000 and $10^4$. In all cases 500 epochs (as in currently reported Table 2) were used.  We observed the good  results for length 1000 and best results for length 2000. Additionally,  we also performed simulations with no subsampling (sample length equal to $4\cdot 10^3$). Then, for $L=3$ or $L=4$ still the best option is to choose $2k$, only for $L=2$ hidden states the best $\texttt{normLL}$ was obtained for 40k (i.e., no subsampling). Details are provided in the following table:
>
> |     |     |     |     |     |     |     |
> | --- | --- | --- | --- | --- | --- | --- |
> | subsampling len: | 500 | 1000 | 2000 | 4000 | 10k | 40k |
> | nr of epochs | 500 | 500 | 500 | 500 | 500 | 500 |
> | L=2 | -0.6309 | -0.9843 | *-0.5619* | -0.5679 | -0.7251 | **-0.256** |
> | L=3 | 0.5409 | 0.5409 | **0.6269** | 0.4729 | 0.4079 | 0.528 |
> | L=4 | 0.7432 | 0.661 | **0.7513** | 0.7387 | 0.515 | 0.7075 |
>
> It is also worth noting that in this case increasing the number of epochs usually improves the results, but still in any case $\mathcal{F}^{\rm{EM}}$ $\texttt{normLL}$s are worse than those of $\mathcal{F}^{\mathbf{Q}}$’s. For example, for L=2 and no sub-sampling we obtained  $\mathcal{F}^{\rm{EM}}$ $\texttt{normLL} = 0.7599$, whereas, as reported in Table 2, $\mathcal{F}^{\mathbf{Q}}$ $\texttt{normLL} = 0.823$. At the same time, for example for $L=2$, training $\mathcal{F}^{\rm{EM}}$  model with no subsampling took 16 hours 22 min, whereas training $\mathcal{F}^{\mathbf{Q}}$ took 4 m 49 s. The differences for $L=3$ or $L=4$ between running times of $\mathcal{F}^{\mathbf{Q}}$ and  $\mathcal{F}^{\rm{EM}}$are of the same order.
>
> We also would like to thank the reviewer for suggesting checking if  $\mathcal{F}^{\rm{EM}}$ would be more competitive with longer sampled sequences and fewer epochs. To this end, we performed additional simulations and trained  $\mathcal{F}^{\rm{EM}}$  without subsampling   for 100, 200, 300 and 500 epochs. The results are provided in the table below (best results in each row **bolded**). As can be seen, in a sense the model is competitive: if e.g., we would always make no subsampling, but instead train the model for 200 epochs (recall, in paper we reported 500 epochs for sub-sampling with length $1k$), then the results ($\texttt{normLLs}$) are always better, the price we pay is longer execution time: for $L=3$ or $L=4$ it is c.a. 5 times larger, whereas for $L=2$ it is c.a. 26 times larger. We are willing to report the results in a revised version of the paper.
>
> |     |     |     |     |     |     |     |
> | --- | --- | --- | --- | --- | --- | --- |
> | sub-sampl len: |     | 1000 | 40k | 40k | 40k | 40k |
> | epochs |     | 500 | 100 | 200 | 300 | 500 |
> | L=2 | $\texttt{normLL}$ | -0.9843 | **-0.08662** | -0.1186 | -0.1615 | -0.256 |
> |     | time | 4m  | 37m | 1h 47m | 2h 32m | 3h 4m |
> | L=3 | $\texttt{normLL}$ | 0.5409 | -0.01105 | **0.6527** | 0.5024 | 0.528 |
> |     | time | 13m | 50m | 1h 17m | 2h 35m | 4h 53m |
> | L=4 | $\texttt{normLL}$| 0.661 | 0.03885 | **0.8006** | 0.7246 | 0.7075 |
> |     | time | 14m | 36m | 1h 18m | 1h 58m | 4h 23m |
>
> Concerning **financial data**:
>
> Yes indeed, usually one works with log returns $\log(y_k/y_{k-1})$ which tend to be “more” Gaussian. Actually we have started with log-returns, all models, i.e.,  $\mathcal{F}^{\rm{EM}}$, $\mathcal{F}^{\mathbf{Q}}$ and Gaussian ones returned almost the same normLLs (they differed really slightly). That is, in a sense, no surprise: Gaussians fitted to Gaussians, the other models also have no difficulty in fitting in such a case. Then we chose (and reported) just differences $y_k - y_{k-1}$ since they are “more non-Gaussian” to show that in such a case  $\mathcal{F}^{\rm{EM}}$ and $\mathcal{F}^{\mathbf{Q}}$ outperform the baseline. In Fig. 6 (in Appendix) we presented the trained distributions (which are non-Gaussian). In the main body of a revised version we intend to mention the reason for choosing differences instead of log-returns.

---

> ### Author Response · Authors · 2022-08-02
> **Official response to reviewer idya (part 2 of 2)**
>
> Concerning the **perturbations during $\mathcal{F}^{\mathbf{Q}}$ training**:
>
> The role of the noise perturbations while training the normalizing flow was widely discussed in response to **Reviewer ujjf**. The value of the Gaussian noise hyperparameter should be selected to cover the area around the grid point, but the probability of mixing perturbed points from various grid points should be minimal. In order to guarantee that we set the variance of the noise equal to $0.1$, which fulfills our requirement in all our examples. We tested the variants with the noise depending on the grid size, but we did not report any difference. Eliminating the noise perturbations caused the distribution spikes for some of the considered examples between the greed points and collapsing of the model during training.
>
> Concerning the **Limitations: Curse of dimensionality**: we suggest using  $\mathcal{F}^{\mathbf{Q}}$ model for low dimensional data (especially with long observation sequences), whereas we suggest using $\mathcal{F}^{\rm{EM}}$ for multi-dimensional data. The latter we also tested on 6 dimensional data. Please, see details in a response to **Reviewer’s ujjf** comment.
> We would like to add few words on a method of grid choice: In initial experiments we tested several methods, focusing later on on three of them: a) we create a uniform grid of size $m$ on each feature; b) for each feature we cluster the data using $k$-means into $m$ clusters, the centroids represent then the grid; c) half of the grid, for given feature, is a result of a), the other half is a result of b).  In any case, the final full grid is of size $m^d$. In all initial experiments “strategy” c) gave the best results, that is why we used it later on in all the examples.
> Of course, we will correct the typos in the revised version.

---

### Official Review · Reviewer_8Abb · 2022-07-08

**Rating:** 5
**Confidence:** 3
**Soundness:** 2 fair
**Presentation:** 3 good
**Contribution:** 2 fair

**Summary:**

The authors propose to use continuous normalizing flows to extend Hidden Markov Models (HMM) by modeling the conditional distribution of the observables given the hidden state with a flow. They call their method FlowHMM and present two training algorithms: First, they point out that the Expectation-Maximization (EM) algorithm can be used for this purpose, which is a common procedure training HMMs. Second, the authors suggest training their FlowHMM model by finding parameters to match the empirically computed co-occurrence matrix of two subsequent observations. They applied their method to several synthetic and real world sequential datasets and thereby demonstrate the effectiveness of the two training algorithms and outperform the baselines given by Gaussians or mixture of Gaussians instead of a continuous normalizing flow modeling the distribution of the observables.

**Questions:**

* Why did you choose continuous normalizing flows as a density estimation method in your model and not another approach such as a discrete normalizing flow or an autoregressive model?

**Limitations:**

The authors highlight in the final section of their paper that the method relies on the co-occurrence matrix for training and is not validated sufficiently on multidimensional time series data.

**Strengths And Weaknesses:**

## Strengths

I appreciate that the article is well organized and easy to read. The authors have an illustrative figure of their procedure, i.e. Figure 1, which ameliorates the understandability of the presented material even further. They also clearly point out their contributions both in the abstract and in the main paper.

Moreover, in their experiment section, the authors consider a variety of datasets, including synthetic once and real-world examples.

## Weaknesses

My main criticism concerns the motivation of the method and its comparison to competing approaches.

The authors chose a continuous normalizing flow to model the conditional distribution of the observables given the latent variables, but this density could have been modeled by any model designed for density estimation, such as a discrete normalizing flow or an autoregressive model. They argue that a continuous normalizing flow is more expressive than a Gaussian or a mixture of Gaussians, which I agree with, but they give no reason why they did not choose another of the advanced density estimation models I mentioned.

Moreover, Ghosh et al. (2020) developed a very similar approach and made a follow-up ([Ghosh et al, 2021](https://arxiv.org/abs/2107.00730)), which is not even mentioned in the paper. They claim that Ghosh's model cannot be applied in the cases they are considering since Real NVP cannot deal with one-dimensional data; however, instead of Real NVP they could just use a Neural Spline Flow or a Residual Flow and made a comparison. This should be done in my point of view to highlight the strength and novelty of the presented method.

## Conclusion

Given that there are essential things missing in the paper as I argued above, I cannot vote for acceptance of this paper at the current stage. I ask the authors to clarify these points and are willing to update my score if they do so.

**Edit**

Since the authors addressed some of my concerns, I'll increase my score slightly.

---

> ### Author Response · Authors · 2022-08-02
> **Official response to reviewer 8Abb (part 1 of 2)**
>
> We thank the Reviewer for the insightful comments about our work, we will respond to them below.
>
> >***Comment:*** Comparison to Ghosh et al. (2021)
>
> We agree with the reviewer that Ghosh et al. (2021) is an important reference that should be included and discussed in our work. The proposed method enriches the paper Ghosh et al. (2020), but the focus is still on a particular application (classification of speech phones), and does not explore the capabilities of the model for other tasks and datasets, including density estimation, hidden states prediction. The direct comparison on 1D data with our approach is difficult, replacing coupling models used by Ghosh et al. (2021) by Neural Spline Flows (Durkan, C. et al. (2019)) or Residual Flows (Chen, R. T et al. 2019) is non-trivial (see the motivation for CNF), and may result in a poor and underrated quality of the reference model.
> Our approach differs significantly from Ghosh et al. (2021). First, the computational complexity of Ghosh et al. (2021) depends on the length of the training sequence, so the application of the model for larger sequences is time-consuming and also may require taking subsequences during training. $\mathcal{F}^{\mathbf{Q}}$ uses a cooccurrence matrix and can be applied on very long sequences. Compared to $\mathcal{F}^{\rm{EM}}$, our approach applies gradient-based optimization not only to the parameters of the flow (as it is done in Ghosh et al. (2021)) but also to $\mathbf{S}$ matrix thanks to the parametrization given by formula (9). Consequently, in each iteration, only one forward step is needed, while Ghosh et al. (2021) model requires two forward passes: with the frozen weights (no grad) and weights for gradient updates.
>
> We apply a comprehensive evaluation of Ghosh et al. (2021) model for 2D examples considered in our experiments, including Examples 5a, 5b, 6a, 6b. In this experiment, we use the official implementation from https://github.com/anubhabghosh/genhmm. Due to time limitations, we applied experiments only for $T=10^3$ and $T=10^4$, but we will provide the complete results in camera-ready if accepted. We will also tune the parameters of the model to our data – initial experiments show that our model outperforms Ghosh et al. (2021) in Example 5a, 6a, 6b, whereas it “loses” in Example 5b. The results are provided in the table below:
>
>
> |     |     |     |     |     |
> | --- | --- | --- | --- | --- |
> |     | paper | rebuttal | paper | rebuttal |
> |     | $\\mathcal{F}^{\\rm{EM}}$ | Ghosh et al. | $\\mathcal{F}^{\\rm{EM}}$ | Ghosh et al. |
> | $T$ | Ex. 5a |     | Ex. 5b |     |
> | $10^3$ | **-0.6003** | -0.7643 | -1.3309 | **-0.743** |
> | $10^4$ | **-0.4822** | -0.7052 | -1.2286 | **-0.7036** |
> |     | Ex. 6a |     | Ex 6b |     |
> | $10^3$ | **1.0531** | 0.1600 | **0.3636** | 0.1487 |
> | $10^4$ | **1.0807** | 0.1673 | **0.3551** | 0.1665 |

---

> > ### Author Response · Authors · 2022-08-02
> > **Official response to reviewer 8Abb (part 2 of 2)**
> >
> > >***Question***: Why did you choose continuous normalizing flows as a density estimation method in your model and not another approach such as a discrete normalizing flow or an autoregressive model?
> >
> > As the Reviewer noticed, in our work, we are focused on modeling distributions of one or low-dimensional data. CNFs were successfully applied in such models as NGGP (Sendera et al. 2021), PointFlow (Yang, G. et. al 2019), or StyleFlow (Abdal, R. et. al 2021), where the dimensionality and characteristic of data are similar. It is somehow confirmed by the empirical results provided by the authors of FFJORD (Grathwohl et al. (2018); Table 2), the proposed approach performs better than discrete flows like RealNVP (Dinh et al. (2016)) or Glow (Kingma & Dhariwal (2018)) for low-dimensional data in terms of normLL. Such models were used in Ghosh et al. (2021). Moreover, flows that use coupling layers (RealNVP (Dinh et al. (2016)), Glow (Kingma & Dhariwal (2018))) and autoregressive flows (MAF (Papamakarios et al. (2017))) do not make sense for 1D data. While operating on 1D data, we do not care about simplifying the estimation of the Jacobian, and any invertible differentiable transformation can be applied. At the same time, we need a complex, well-parameterized transformation that is delivered by a dynamic function of CNF.
> >
> > Considering Neural Spline Flows (Durkan, C. et al. (2019)), they represent a good alternative for CNF models considering low-dimensional data, which was confirmed on standard benchmark datasets. However, this model was not tested on 1D data, and using the original implementation of that model assumes incorporating coupling layers that are designed for higher dimensions. It is not trivial how to adjust the model to the one-dimensional case using a well-parametrized neural network. Moreover, the model requires tuning of additional hyperparameters like a number of bins. Residual Flows (Chen, R. T et al. 2019) are rather focused on high-dimensional image data, where the authors propose to overcome the problem of calculating the determinant of the Jacobian.
> >
> > In conclusion, the CNFs are the best choice considering adaptation to 1D data, quality for low-dimensional distribution modeling, and hyperparameter calibration issues.
> >
> > - Damianou, A., & Lawrence, N. D. (2013, April). Deep Gaussian Processes. In Artificial intelligence and statistics (pp. 207-215). PMLR.
> > - Dinh, L., Sohl-Dickstein, J., & Bengio, S. (2016). Density estimation using Real NVP. arXiv preprint arXiv:1605.08803.
> > - Grathwohl, W., Chen, R. T., Bettencourt, J., Sutskever, I., & Duvenaud, D. (2018). Ffjord: Free-form Continuous Dynamics for Scalable Reversible Generative Models. arXiv preprint arXiv:1810.01367.
> > - Kingma, D. P., & Dhariwal, P. (2018). Glow: Generative Flow with Invertible 1x1 Convolutions. arXiv preprint arXiv:1807.03039.
> > - Papamakarios, G., Pavlakou, T., & Murray, I. (2017). Masked Autoregressive Flow for Density Estimation. arXiv preprint arXiv:1705.07057.
> > - Ghosh, A., Honoré, A., Liu, D., Henter, G. E., & Chatterjee, S. (2021). Normalizing flow based hidden Markov models for classification of speech phones with explainability. arXiv preprint arXiv:2107.00730.
> > - Sendera, M., Tabor, J., Nowak, A., Bedychaj, A., Patacchiola, M., Trzcinski, T., & Zieba, M. (2021). Non-Gaussian Gaussian Processes for Few-Shot Regression. Advances in Neural Information Processing Systems, 34, 10285-10298.
> > - Yang, G., Huang, X., Hao, Z., Liu, M. Y., Belongie, S., & Hariharan, B. (2019). Pointflow: 3d point cloud generation with continuous normalizing flows. In Proceedings of the IEEE/CVF International Conference on Computer Vision (pp. 4541-4550).
> > - Abdal, R., Zhu, P., Mitra, N. J., & Wonka, P. (2021). Styleflow: Attribute-conditioned exploration of stylegan-generated images using conditional continuous normalizing flows. ACM Transactions on Graphics (ToG), 40(3), 1-21.
> > - Durkan, C., Bekasov, A., Murray, I., & Papamakarios, G. (2019). Neural spline flows. Advances in neural information processing systems, 32.
> > - Chen, R. T., Behrmann, J., Duvenaud, D. K., & Jacobsen, J. H. (2019). Residual flows for invertible generative modeling. Advances in Neural Information Processing Systems, 32.
> >
> > Concerning the **Limitations:**: we suggest using  $\mathcal{F}^{\mathbf{Q}}$ model for low dimensional data (especially with long observation sequences), whereas we suggest using $\mathcal{F}^{\rm{EM}}$ for multi-dimensional data. The latter we also tested on 6 dimensional data. Please, see details in a response to **Reviewer’s ujjf** comment.

---

> > > ### Comment · Reviewer_8Abb · 2022-08-09
> > > **Upgrade of my score**
> > >
> > > Thanks for addressing some of my concerns. I still think there are discrete flows which could work equally well in this setting, but I acknowledge the effort of adding an additional baseline and update my score slightly.

---

### Official Review · Reviewer_eTi7 · 2022-07-11

**Rating:** 8
**Confidence:** 4
**Soundness:** 4 excellent
**Presentation:** 4 excellent
**Contribution:** 4 excellent

**Summary:**

This paper introduces FlowHMM for modeling time-series data. The authors explain that HMMs can either be trained using EM, which is slow and prone to being in poor local minima, or using a co-ocurrence matrix in the case of discrete HMMs. Additionally, HMMs model the observations via simple, tractable distributions like Mixtures of Gaussians. Instead, FlowHMM models these observations using normalizing flows.

The authors propose two techniques to train their model. The first is with the standard EM approach, and the second approach involves discretizing the continuous HMM into a discrete one, and then using a co-ocurrence matrix based approach.


**Questions:**

Can you provide some insight into how to decide on which training technique to use? (EM, or discretization and co-occurrence matrix). Specifically, can you show empirically what happens when you use co-occurrence matrix when “the assumption that the Markov chain on a set of hidden states is stationary - a limitation not present in the EM-based approaches” (line 330) does not hold.


**Limitations:**

I do not see any potential negative societal impact.

**Strengths And Weaknesses:**

The authors very carefully describe how they can exploit the structure of their FlowHMM model (flow-based emission probabilities) to easily get a good co-occurence matrix based approach for training their model. I feel that this should be highlighted at the end of the introduction.

---

> ### Author Response · Authors · 2022-08-02
> **Official response to reviewer eTi7**
>
> We thank the Reviewer for the insightful comments about our work, we will respond to them below.
> >***Question:*** Can you provide some insight into how to decide on which training technique to use? (EM, or discretization and co-occurrence matrix). Specifically, can you show empirically what happens when you use co-occurrence matrix when “the assumption that the Markov chain on a set of hidden states is stationary - a limitation not present in the EM-based approaches” (line 330) does not hold.
>
> Concerning *when to use each model ($\mathcal{F}^{\rm{EM}}$ or $\mathcal{F}^{\mathbf{Q}}$)*:
>
> We had following situation in mind   for lower-dimensional models we mainly use $\mathcal{F}^{\mathbf{Q}}$, whereas for high-dimensional model we mainly use $\mathcal{F}^{\rm{EM}}$.  Each has its advantages and disadvantages. Please, see a detailed argumentation provided in a response to **Reviewer ujjf** comment.
>
> Concerning *stationarity of a Markov chain*:
>
> Note that we consider only Markov chains with at most $L=3$ states. In such a case, even if the assumption of stationarity does not hold, it does not have a large impact on the results. For finite state-space Markov chains, the rate of convergence to stationarity is exponential (in number of states), which in practice means that for the aforementioned 3 states only few steps are needed to be very close to stationarity. To be more specific, for the chain with transition matrix A_2 given in (17), the stationary distribution is $\pi(1)=5/14, \pi(2)=4/14, \pi(3)=5/14$. One can check, that e.g., after 10 steps the total variation between the stationary distribution and the distribution of $X_{10}$ is at most (for any initial distribution) $4.21775 \cdot 10^{-6}$.

---

> > ### Comment · Reviewer_eTi7 · 2022-08-04
> > **Response**
> >
> > Thank you for your response. You have thoroughly answered my questions here, and in the responses to the other reviews. As a result, I've increased my score.

---

### Official Review · Reviewer_ujjf · 2022-07-12

**Rating:** 4
**Confidence:** 4
**Soundness:** 3 good
**Presentation:** 3 good
**Contribution:** 2 fair

**Summary:**

The paper presents the FlowHMM, a novel combination of a hidden Markov models (HMMs) and flow based deep network architectures, along with two methods of training the model. The first method is based on expectation-maximization (EM), and the second is based on matching co-occurrence matrices computed from data. Experiments are performed comparing both methods with a Gaussian mixture model baseline.

**Questions:**

Some typos I found:
- Line 142: I believe it should be $\sum_{v_i,v_j\in\mathcal{V}}$
- Line 180: "due to the fact"

Questions:
- In equation (14) you treat the matrices $Q_\beta$ and $\hat{Q}$ as probabilistic distributions. Given how these matrices are defined, is there a way to make this more rigorous (i.e. give a probabilistic interpretation of $Q_\beta$ and $\hat{Q}$ in their definition)?
- In equation (16) random perturbations are added to the grid values for the computation of $P_\beta$ to prevent overfitting. Is there a reason why the same is not done for $\hat{Q}$ in equation (15)?
- Have you tried discretization for higher dimensional data? Are the results competitive with mixture models/other flow-based models (Ghosh 2020)?
- Could you clarify what you mean by "as a black box" (line 248) and "discover the hidden states of the data unobserved during training" (line 250)? Are you suggesting that it is hard to do inference in the model by Ghosh et. al.?

**Limitations:**

As I mentioned in the strengths and weaknesses section, the main limitation I see with this work is the difficultly of learning in higher dimensions, which the authors do not address in the main paper.

**Strengths And Weaknesses:**

The main ideas of the paper are clearly presented, and the details of the model along with two different training schemes are treated with care. The background on how continuous HMM models are trained in the past and their pros and cons are also made very clear to the reader. The experiment section provides comprehensive results on fitting several low-dimensional datasets, including a total of 6 different datasets/variants, including 2 real datasets and 2 synthetic ones.

Despite the aforementioned strengths, however, the paper has a couple of weaknesses that are hard to overlook. One major weakness is the lack of comparison with similar continuous non-linear HMM models (e.g. structured variational autoencoders with HMM dynamics). Only comparing with Gaussian mixture HMMs doesn't seem to be sufficient and renders the results somewhat weak in my opinion.

Furthermore, the $\mathcal{F}^Q$ model, which seems to be the more novel of the two presented, yields unsatisfactory results in the majority of runs in examples 5 and 6 (table 3), which featured 2 dimensional datasets. This huge drop of performance is not addressed or explained in the paper.

To me, it seems like the drop of performance for $\mathcal{F}^Q$ in 2D is a consequence of the discretization of the observation space that is central to the approach. As dimensionality increases the discretization is going to incur an exponential cost which seems highly problematic. This point is breifly addressed on line 208 but I believe it deserves more attention and perhaps experimental verification.

Lastly, it is claimed that the EM training method is inefficient for long sequences, but the computational costs are not reported or analyzed. A suggestion would be to talk about these computational tradeoffs to give more information about the practical value of these algorithms.

To conclude, the paper is pretty well written and the model proposed is somewhat novel, but the experimental results are quite weak and the difficultly of fitting to high-dimensional data is an important problem that is missing from this work.

---

> ### Author Response · Authors · 2022-08-02
> **Official response to reviewer ujjf  (part 1 of 3)**
>
> We thank the Reviewer for the insightful comments about our work, we will respond to them below.
>
> >***Comment:*** “… lack of comparison with similar continuous non-linear HMM models (e.g. structured variational autoencoders with HMM dynamics).
>
> In our experimental evaluation, we are mainly focused on one or low-dimensional datasets. A mixture of Gaussians with a sufficient number of components should approximate any distribution well. According to [1]: *"A Gaussian mixture model is a universal approximator of densities, in the sense that any smooth density can be approximated with any speciﬁc nonzero amount of error by a Gaussian mixture model with enough components"*. Note however, that the number of Gaussian components in the mixture usually must be large to approximate some densities. In a submitted paper we encourage the reader to compare our Fig. 2 with [4, Fig. 4] (in the submitted version this is ref. [7, Fig. 4]). The authors therein approximate a uniform distribution – they needed 123 components to achieve their accuracy. The same uniform distribution is estimated *“with less effort”* via the flow model.
>
> Models like SVAE [2, 3] are rather designed for data characterized by higher dimensions, where the VAE bottleneck is well motivated. The comprehensive evaluation of such approaches is limited, because such models provide ELBO, not direct $\texttt{normLL}$ calculation. It can be achieved by the application of the flow component FlowHMM.
>
> Moreover, we have used the official implementation of Ghosh et al. (2021) and applied it to our synthetic 2D datasets used in Examples 5a-b, 6a-b – initial experiments show that our model outperforms it in Examples 5a, 6a and 6b. Please, see the response to a comment of **Reviewer 8Abb** for details, including the Table with values of $\texttt{normLL}$s.
>
>
> - [1] Goodfellow, I., Bengio, Y., & Courville, A. (2016). Deep learning. MIT press.
> - [2] Johnson, M. J., Duvenaud, D. K., Wiltschko, A., Adams, R. P., & Datta, S. R. (2016). Composing graphical models with neural networks for structured representations and fast inference. Advances in neural information processing systems, 29.
> - [3] Ebbers, J., Heymann, J., Drude, L., Glarner, T., Haeb-Umbach, R., & Raj, B. (2017, August). Hidden Markov Model Variational Autoencoder for Acoustic Unit Discovery. In InterSpeech (pp. 488-492).
> - [4] Balaji Lakshminarayanan and Raviv Raich. Non-negative matrix factorization for parameter
> estimation in hidden markov models. In 2010 IEEE International Workshop on Machine
> Learning for Signal Processing, pages 89–94, 2010.
>
>
>
> >***Comment:*** The drop of quality for the $\mathcal{F}^{\mathbf{Q}}$ model for 2D data should be discussed.
>
> Indeed we should have remarked: For 2D datasets we can spot a drop of quality for $\mathcal{F}^{\mathbf{Q}}$ model. However, for Example 6 (both a and b) and longer observation sequences ($T\geq 10^4$ ) the performance measured in accuracy is quite comparable to other models (and in one case it yields the best result). For all the examples (i.e., 5a-b, 6-ab) and $T=10^3$ we performed additional simulations: we trained model $\mathcal{F}^{\mathbf{Q}}$ using grid of size $M=35$ and $M=40$. For Example 5a we obtained  $\texttt{normLL}$ $ -2.548$ for $M=35$ and $-2.4999$ for $M=40$. Recall, in paper we considered $M=30$ obtaining $normLL$ -2.709. Exactly the same phenomena (i.e., increasing $M$ increases $\texttt{normLL}$) holds for Example 5b, 6a-b (details in table below). It shows that increasing the grid size increases the performance of the model. Note that in this ($T=10^3$) case the $\mathcal{F}^{\mathbf{Q}}$ result are worse than $\mathcal{F}^{\rm{EM}}$ (with $\texttt{normLL}$ being $ -0.6003$), which is no surprise: there are $30^2 (35^2, 40^2)$ pairs for $M=30$ ($M=35, M=40$), whereas there are only $10^3$ observations.
>
>
> |     |     |     |     |
> | --- | --- | --- | --- |
> |     |  $\mathcal{F}^{\mathbf{Q}} ,M=30$|  $\mathcal{F}^{\mathbf{Q}}, M=35$ |  $\mathcal{F}^{\mathbf{Q}}, M=40$ |
> | 5a  | -2.709 | -2.548 | -2.499 |
> | 5b  | -2.679 | -2.52 | -2.47 |
> | 6a  | -2.182 | -2.04 | -1.997 |
> | 6b  | -2.189 | -2.042 | -1.99 |

---

> > ### Author Response · Authors · 2022-08-02
> > **Official response to reviewer ujjf  (part 2 of 3)**
> >
> > >***Comment:*** “EM training method is inefficient for long sequences, but the computational costs are not reported or analyzed. A suggestion would be to talk about these computational tradeoffs to give more information about the practical value of these algorithms.”
> >
> > In Appendix (sec. A.6) in Table 4, we report the average time per epoch, for different lengths of training sequences, for both training approaches. For long sequences we randomly sampled subsequences (in each step of the EM training method) of length $10^3$, as reported in Appendix A.6. Additionally, we have performed exhaustive simulations for “Air pressure” (Example 4) dataset (where the training observation was of length 40k), where we consider EM training with subsampling a subsequence at each step with different lengths, as well as no subsampling. We consider the results ($\texttt{normLL}$s) and execution time – please, see the response to Reviewer idya   comments.
> >
> >
> >
> >
> >
> >
> >
> >
> >
> > >***Question:*** In equation (14) you treat the matrices $\mathbf{Q}_\boldsymbol{\beta}$ and $\hat{\mathbf{Q}}$ as probabilistic distributions. Given how these matrices are defined, is there a way to make this more rigorous (i.e. give a probabilistic interpretation of $\mathbf{Q}_\boldsymbol{\beta}$ and $\hat{\mathbf{Q}}$  in their definition)?
> >
> > Both $\hat{\mathbf{Q}}$  and $\mathbf{Q}\boldsymbol{\beta}$  matrices can be interpreted as categorical distributions. The $(i,j)$-th entry of the matrix  represents  the probability of observing a pair of states (v_i, v_j) at some fixed two consecutive time steps $t$ and $t+1$. This is independent from $t$, since throughout the paper we assume that the underlying Markov chain on hidden states is stationary (thus bivariate distribution of $(X_t, X_{t+1})$ is independent from $t$. $\hat{\mathbf{Q}}$ represents the ML estimator of such probabilities given by eq. (10), calculated using the training sequence. $\mathbf{Q}_\boldsymbol{\beta}$ represents the probabilities returned by $\mathcal{F}^{\mathbf{Q}}$ model that are calculated according to formula (9). The fact that $\mathbf{Q}_\boldsymbol{\beta}$ is indeed a distribution was provided in line 142. We will elaborate on the interpretation of both matrices in the revised version of the manuscript
> >
> >
> > >***Question:*** In equation (16), random perturbations are added to the grid values for the computation of $\mathbf{O}_\boldsymbol{\beta}$ to prevent overfitting. Is there a reason why the same is not done for $\hat{\mathbf{Q}}$  in equation (15)?
> >
> > The generative models aim at approximating true data distribution, but practically we do not have access to that distribution but to the limited data represented by the train set. This limitation is especially painful for models that optimize directly $\texttt{normLL}$, like normalizing flows. In order to reduce the negative impact of this phenomenon, the grid points are perturbed in each training iteration with relatively small Gaussian noise. This process imitates the situation where we have access to an infinite number of training examples and, as a consequence, reduces the negative effect of overfitting and collapsing. In other words, adding the noise covers the spaces around the grid points, and smooths the approximated density function. Applying this procedure to $\hat{\mathbf{Q}}$  is an interesting direction (equivalent to dynamic prior for categorical distribution), but perturbing the examples processed by the flow directly imitates the situation, where the flow observes slightly different examples in each iteration and is consistent with good practices, while training such a models.

---

> > > ### Author Response · Authors · 2022-08-02
> > > **Official response to reviewer ujjf  (part 3 of 3)**
> > >
> > > >***Question***: Have you tried discretization for higher dimensional data? Are the results competitive with mixture models/other flow-based models (Ghosh 2020)?
> > >
> > > We had following situation in mind (and probably should have remarked it clearer – somewhere around lines 207-208): for lower-dimensional models we mainly use $\mathcal{F}^{\mathbf{Q}}$, whereas for high-dimensional model we mainly use $\mathcal{F}^{\rm{EM}}$.  Each has its advantages and disadvantages, main ones are: $\mathcal{F}^{\mathbf{Q}}$ is well-suited for lower-dimensional data with long observation sequence (recall, we compute matrix $\mathbf{Q}$ once), in such a case $\mathcal{F}^{\rm{EM}}$ will be very slow (each epoch loops through whole observation sequence).  We have one example with a long observation sequence – “Air pressure” data (Example 4, with the train set of length $T=40k$), where to shorten the execution time we randomly sampled at each step of $\mathcal{F}^{\rm{EM}}$  approach a subsequence of length $10^3$. There is a trade-off between execution time and performance – see the table in a response to Reviewer idya.  On the other hand, for higher-dimensional data the discretization may be cumbersome, that is why we suggest using $\mathcal{F}^{\rm{EM}}$  then.
> > >
> > > We have performed additional simulations  for a more-dimensional real dataset. To be more exact, we considered  AmeriGEOSS data [24] (the same source as “Air pressure”), from which we have chosen 6 features related to  humidity (mean, stdev) and air pressure (min, max, mean, stdev). Similarly as in Example 4, we considered differences of observations. Again, the length of the training set was set to $40k$, the remaining (${\sim}14k$) observations constituted the test set. In the spirit of what we described above – we used the $\mathcal{F}^{\rm{EM}}$ model. This model significantly outperforms Gaussian baselines in all the cases ($L=2,3,4$). Roughly speaking, $\texttt{normLL}$ is about $-1000$ for Gaussian models, whereas it is around $-570$ for our model. Details are provided in the table below.
> > >
> > > |     |     |     |     |     |
> > > | --- | --- | --- | --- | --- |
> > > | $L$   | $\mathcal{G}$   | $\mathcal{G}^{10}$ | $\mathcal{G}^{20}$ |$\mathcal{F}^{\rm{EM}}$ |
> > > | 2   | -1097.49 | -1096.81 | -1096.65 | -571.08 |
> > > | 3   | -1097.45 | -1096.73 | -1096.51 | -571.97 |
> > > | 4   | -1097.39 | -1096.65 | -1096.09 | -572.61 |
> > >
> > > >***Question:*** Could you clarify what you mean by "as a black box" (line 248) and "discover the hidden states of the data unobserved during training" (line 250)? Are you suggesting that it is hard to do inference in the model by Ghosh et. al.?
> > >
> > > To clarify, for the model proposed by Ghosh et. al. it is possible to infer and predict the hidden states, but the authors applied it only as a generative model for sequence classification, they do not analyze the clustering capabilities of the model, and how the proposed approach disentangles the mixture of sequences.
> > >
> > >  Of course we will correct the typos (e.g., those in lines 142 and 180) in a revised version.

---

> > > > ### Comment · Reviewer_ujjf · 2022-08-09
> > > > **Increasing my score**
> > > >
> > > > Thank you for the in-depth response which was very helpful in clearing some of the confusions I had. I think the authors have done a good job on the specific setting of one dimensional datasets, which is the main focus of this work. Futhermore, experiment results are shown for higher dimensional datasets, which directly address the curse of dimensionality concern. Upon further consideration I think my initial reviews are overly harsh and have increased my overall score from 3 to 4.

---

### Official Review · Reviewer_5RVg · 2022-07-12

**Rating:** 7
**Confidence:** 3
**Soundness:** 3 good
**Presentation:** 4 excellent
**Contribution:** 3 good

**Summary:**

The paper proposes a method that utilizes NFs to construct a more general distribution over observations.
The paper provides two different training methods based on the EM algorithm or the co-occurence method.
The paper demonstrates the efficacy of both methods on synethetic and real datasets.

**Questions:**

See above

**Strengths And Weaknesses:**

Originality: Are the tasks or methods new? Is the work a novel combination of well-known techniques? (This can be valuable!) Is it clear how this work differs from previous contributions? Is related work adequately cited

Nothing presented is particularly novel in its own right but, to my knowledge, the combination of ideas is novel and the paper explores the combinations thoroughly (though additional baselines would have been nice wrt the real datasets).


Quality: Is the submission technically sound? Are claims well supported (e.g., by theoretical analysis or experimental results)? Are the methods used appropriate? Is this a complete piece of work or work in progress? Are the authors careful and honest about evaluating both the strengths and weaknesses of their work?

The paper appears to be technically sound and the results are consistent with the claims. Current limitations and directions for future work are (briefly) discussed. These directions seem reasonable as a separate work and I do not think they need to be included in this effort.


Clarity: Is the submission clearly written? Is it well organized? (If not, please make constructive suggestions for improving its clarity.) Does it adequately inform the reader? (Note that a superbly written paper provides enough information for an expert reader to reproduce its results.)

The paper is well-written and well-cited. I have no major complaints. There are a few minor typos (e.g., line 180 "...due *ti* the fact..."). The figures and synthetic experiments do an excellent job of demonstrating and clarify the ideas discussed in the text.


General comments:

Overall, I found this paper very accessible and the proposed method intuitive. The paper presents its ideas clearly and demonstrates them reasonably well. It would have been nice to see additionaly baselines on the real datasets and might have been nice to see higher dimension examples.

---

> ### Author Response · Authors · 2022-08-02
> **Official response to reviewer 5RVg**
>
> We thank the reviewer for her/his positive feedback. We would like to point out that the reviewer acknowledges the clarity of our paper and its accessibility. Regarding the additional baselines and higher dimensional examples: a) We have added several experiments on higher dimensionality real data (where the proposed methods remain superior to the other approaches); b) we used Ghosh et al. (2021) implementation to compute $\texttt{normLL}$s for our 2D examples (Example 5a-b, 6a-b) (where the proposed methods remain superior in most cases). Please, see the details in  comments to other reviewers).

---

### Author Response · Authors · 2022-08-02
**Official comments to Area Chairs and Reviewers**

We thank the Reviewers for their helpful comments. The reviewers appreciate the contribution and novelty of the work. The primary concern is the empirical evaluation of the proposed method for high-dimensional data. As we expressed in the article, we propose two training strategies for our model, where $\mathcal{F}^{\mathbf{Q}}$ is designed for long, one-dimensional sequences, and $\mathcal{F}^{\mathbf{EM}}$ is proposed for higher dimensions. As a response to reviews, we evaluated $\mathcal{F}^{\mathbf{EM}}$ on 6-dimensional datasets and compared it against selected baselines.

Reviewer **8Abb** had some concerns regarding the comparison to Ghosh et al. (2021) and the motivation behind selecting CNF. We elaborated on that, deeply motivated the selection of such flow type, and pointed out the differences between those two approaches. Moreover, we performed empirical studies using Ghosh et al. (2021) as a baseline model for each considered multidimensional example, and $\mathcal{F}^{\mathbf{EM}}$ achieved better or competitive results in most cases. Following the suggestions of the reviewer **idya** we also applied extensive ablation studies for various  lengths of sampled  subsequences (for long sequences of observations, our $\mathcal{F}^{\mathbf{EM}}$ by default samples in each epoch a subsequence of observations of fixed length)  regarding the computational costs remarks. Finally, we commit ourselves to correct all typos and notation remarks and also include extended empirical results for Ghosh et al. (2021) and multidimensional examples in the camera-ready if accepted.

Let us summarize **additional experiments** that we performed for this rebuttal:

- extending Example 4 (real data, 1D, air pressure dataset, training set of length 40k)
	- by default our $\mathcal{F}^{\rm{EM}}$ model in each epoch samples a subsequence of observations of length $1k$. In addition we performed the simulations with lengths $2k, 4k, 10k, 40k$ (the last one actually means 'no subsampling'). All cases with 500 epochs
	- For no subsampling case (equiv. to subsampling with length $40$) following the suggestion of one of the reviewers, we additionally experimented with fewer, 100, 200 and 300, epochs.
- extending 2D Examples 5a-b, 6a-b
	- we performed the simulations for grid size 35 and 40 (in addition to 30 reported in the paper) for the $\mathcal{F}^{\mathbf{Q}}$  model.
	- we adjusted Ghosh et al. (2021) implementation and applied it to our data for $L=3$ hidden states and train sets of lengths $T=10^3$ and $T=10^4$ (due to time limitations during rebuttal period, we postponed $T=10^5$ for later). We compare this model with our models.
- new higher-dimensional example.
	- From AmeriGEOSS dataset [24] (from which we earlier took 1D "Air pressure" data) we have chosen 6D data (training set of length 40k, test set of length ${\sim}14k$) and applied our $\mathcal{F}^{\rm{EM}}$ which resulted in much better performance than  Gaussian baselines.

---

### Meta-Review · Area_Chair_BfvY · 2022-08-25

**Recommendation:** Accept
**Confidence:** Certain

**Metareview:**

the paper proposes a method that combines a hidden Markov model with neural flow models for modelling sequences. The main contribution is a fitting method based on a combination of gradient-based learning and Expectation Maximization, with variants of the method proposed for different data scenarios.

The reviewers found that the paper was very strong from a clarity and quality standpoint. There were some small questions about the novelty - the building blocks of FlowHMM are a HMM and a neural flow model, but the reviewers and I agree that his is an interesting combination and that the inference methods are interesting and novel.

The work is well investigated. The lowest-scoring reviewer (ujjf ) requested more experimentation, which the authors provided wholeheartedly in the rebuttal. Reviewer ujjf upped their score from 3 to 4 based on this, which I'd consider still harsh given the extent of the experimentation in the paper and in the rebuttal.

Overall, the reviewers agree that his is a compelling paper.

**Award:**

No

---

### Decision · Program_Chairs · 2022-09-14

Accept